# Comparative transcriptomics reveal the common anteroposterior molecular blueprint of adult bilaterian guts

Stefano Davide Vianello[1¤]*, Ching-Yi Lin[2], Wahyu Cristine Pinem[2,3], Han-Ru Li[2], Kun-Lung Li[2], Grace Sonia[2], Shu-Hua Lee[1], Szu-Kai Wu[1], Vincent Laudet[1,4,5]*, Yi-Hsien Su[2]*, Jr-Kai Yu[1,2]*, Stephan Q. Schneider[2,3]*

1 Marine Research Station, Institute of Cellular and Organismic Biology, Academia Sinica, Yilan, Taiwan, 2 Institute of Cellular and Organismic Biology, Academia Sinica, Taipei, Taiwan, 3 Department of Bioscience and Biotechnology, National Taiwan Ocean University, Keelung, Taiwan, 4 Marine Eco-Evo-Devo Unit, Okinawa Institute of Science and Technology Graduate University, Onna son, Okinawa, Japan, 5 CNRS IRL 2028 "Eco-Evo-Devo of Coral Reef Fish Life Cycle" (EARLY), Okinawa Institute of Science and Technology Graduate University, Onna son, Okinawa, Japan

¤ Current address: National Institute of Molecular Biology and Biotechnology, National Science Complex, University of the Philippines Diliman, Quezon City, Philippines
* stefano@vianello.ai (SDV); vincent.laudet@oist.jp (VL); yhsu@as.edu.tw (Y-HS); jkyu@gate.sinica.edu.tw (J-KY); sqschneider@gate.sinica.edu.tw (SQS)

## Abstract

A through-gut is one of the major features of bilaterians. Comparative work among bilaterians has identified common molecular mechanisms during early gut patterning, but the primordial gut later often undergoes different degrees of reorganization in each lineage to form a fully differentiated adult gut with specializations along its anteroposterior (AP) axis. Yet, how the conserved embryonic gut AP pattern relates to the adult guts in diverse bilaterians after metamorphosis is still poorly understood. To unravel the molecular subdivisions of adult guts, we investigated the gut through transcriptomic analyses of five phylogenetically informative species: an annelid, a sea urchin, a hemichordate, a cephalochordate, and a vertebrate. We identified bipartite transcriptional programs defining the AP functional subdivisions. Patterning systems composed of Hox, ParaHox, and, surprisingly, other transcription factors (TFs) known to be involved in gut formation in sea urchin larvae are maintained in these adult tissues. Using unbiased analyses, we identified five conserved TF modules corresponding to the AP compartments of the guts that are elaborated or shifted in different species. Our study inferred conserved and modified adult AP patterning modules along bilaterian guts enabling the reconstruction of ancestral bilaterian features with profound implications for the evolution of the bilaterian body plan.

**Data availability statement:** Raw reads of bulkRNAseq datasets are available to download under NCBI BioProject accession number PRJNA1276984 : "Comparative transcriptomics analysis of the anteroposterior organisation of adult bilaterian guts." Counts matrix and code to reproduce the transcriptomics analysis is available at: https://github.com/StefanoVianello/ASICOB_GCP_AdultBilaterianGuts, archived at https://doi.org/10.5281/zenodo.17746910.

**Funding:** This work was funded by Academia Sinica's Grand Challenge grant number AS-GC-111-L01 ("Towards an integrated understanding of metamorphosis in bilaterians"; https://ror.org/05bxb3784) awarded jointly to VL, YHS, JKY, SQS. The funder played no role in the study design, data collection and analysis, decision to publish, or preparation of the manuscript.

**Abbreviations:** AP, anteroposterior; CPM, counts per million; IACUC, Institutional Animal Care and Use Committee; ICOB, Institute of Cellular and Organismic Biology; MGSA, model-based gene set analysis; sPLS-DA, Partial Least Squares-Discriminant Analysis; TF, transcription factor.

## Introduction

### The through-gut: An evolutionary old invention in bilaterians

A through-gut, a "one-way" digestive organ with a mouth and anus, is one of the major features of animals with bilateral symmetry, the bilaterians. Vertebrates, including humans, share this feature with, e.g., sea urchins, acorn worms, flies, tardigrades, mollusks, and worms, suggesting that an inner digestive tube was already established in their last common ancestor.

Through-guts have, however, also been identified within non-bilaterian clades and phyla which are not currently believed to have had a through-gutted ancestor [1–4]. Within the Cnidaria, which classically harbor instead a blind-gut configuration in which food enters and exits from a single, shared external opening [5], some species do indeed display veritable "one-way" through-guts [6], or guts that are not completely blind due to mixed anatomical solutions [7]. Another non-bilaterian phylum, Ctenophora (comb jellies), has also long been known to possess through-guts [8–10].

The actual homology between these non-bilaterians and bilaterian through-guts remains uncertain. Such an assessment is mostly complicated by the outstanding phylogenetic and paleontological uncertainty on the order of emergence of metazoan phyla [11–15], and the consequent debated phylogenetic position of key non-bilaterian and bilaterian phyla with divergent gut architectures [16–19]. Even within extant species, assessments of homology between through-guts are still restricted by a relative limitedness of histological/molecular/genetic characterizations of bilaterian and non-bilaterian key model species. Though efforts to find a shared comparative basis between animal guts are making significant progress in this area (from a developmental biology perspective, see below), there remains a fundamental uncertainty about which regions of the gut should be used as terms of comparison between adult guts with different architectures, or even between through-guts of different species.

Accordingly, even though some bilaterians do not have a through-gut (most notably, many Platyzoa; [20]), the prevalence of through-guts across bilaterians [20,21], the current rooting of both Cnidarians and Ctenophores to blind-gut ancestors [1–4], and the recent identification of through-gutted Ediacaran fossils with bilaterian affinity [22,23] currently do support the interpretation of through-gut evolution as a milestone developmental transition at the base of the bilaterian clade rather than a repeatedly lost earlier metazoan innovation.

### Functional and anatomical divergence of a common bilaterian through-gut

Today, bilaterian adult through-guts display a remarkable variety of forms and functions across phyla, species, and diets [24–27]. Beyond the widespread variation in gut length and/or gut cell type proportions familiarly associated with dietary specializations [28,29], examples include veritable anatomical oddities such as the "fractal" gut anatomy of branched annelids [30,31], and highly divergent anatomical adaptations such as the "armored" stomach of scorpion-eating mice [32]. Similarly, some bilaterian through-guts are capable of functions that have generally found few or no equivalents in other species: the through-gut of many holothurians can be everted as

a defense mechanism [33], and the gut of two recently extinct frog species was uniquely able to temporarily dedifferentiate into a non-digestive brooding chamber [34]. Examples of through-gut anatomical and functional variation abound, and add to the parallel remarkable variety in presence, number, and form of buds/diverticula/glands/organs of unclear homology that connect to or branch off from varying points of the gut tube across many bilaterian species, or seem to be completely missing instead in others [35].

Yet, even in the face of such diversity, a picture starts to emerge whereby the remarkable variety of bilaterian through-gut forms and functions may be understood as the result of (divergent or convergent) evolutionary, ontogenetic, or ecological modification of an underlying common structure. That is, the placement of the through-gut at the base of the bilaterian tree predicts that an underlying set of core anatomical and/or functional modules would exist and be homologous across all bilaterian through-guts.

Current investigative efforts in this direction are mostly searching for such homologies in terms of conserved gene expression patterns and conserved gene regulatory networks: molecular data assessed at phylotypic stages, at early embryonic/larval stages where the through-gut shows the first signs of patterning, and at gastrulation [36–41]. The formation of the embryonic basis that will act as a substrate for the elaboration of postembryonic identities (e.g., in vertebrates, the embryonic gut tube) is itself highly divergent and non-equivalent across species: bilaterian through-guts are formed through a wide variety of embryonic processes, germ layer contributions, and non-equivalent topological transformations [40,42–45]. Yet, investigations of cross-species correspondences in developing guts show strong conservation of antero-posterior (AP) patterns of gene expression and of the underlying gene-regulatory networks during the patterning process itself, even across distant phyla [36–38,46–48]. Such deep conservation of molecular signatures, developmental mechanisms, and patterning systems makes the independent repeated evolution of bilaterian through-guts unlikely, and has allowed to sketch out broad homologies between embryonic gut territories [36,39]. The diversity of bilaterian guts would have, at least in these terms, a common, comparable basis, further elaborated upon by gains, losses, and rearrangements throughout the bilaterian tree. Critically, investigations based on gene expression signatures also allowed to draw new, non-intuitive connections even between highly divergent gut structures, further allowing the discrimination between ancestral and derived functions. Most recently, the link between the vertebrate pancreas and specific cell types and cell functions was found within the gut tube of sea urchin larvae [49,50]. Similar considerations apply to the vertebrate liver organ, though the evolutionary rooting of this organ has up to now focused on the chordate branch [51] and the homology with protostome liver-like organs/cells is still to be assessed [35,52–54].

In general, much is left in being able to distinguish between true through-gut innovations and highly divergent yet homologous structures. It also fundamentally remains unclear whether correspondences between through-gut regions may extend beyond the tripartite "foregut," "midgut," "hindgut" framework that has been defined from embryonic and larval gene expression patterns. Critically, identifying a shared signature across extant bilaterian guts, conserved across millions of years of evolution, may provide a possible identikit of the likely features of its ancestral counterpart. Conversely, the modes through which the original through-gut might have been used to interact with, process, and exploit a wide variety of nutrients, and their effective distribution across its AP length, may have left a deep homologous signature to be recovered across extant bilaterian through-guts, regardless of their current digestive solutions and highly divergent gastrointestinal variations.

## Justification of current study

In this study, we analyze anteroposteriorly-resolved transcriptomic data collected along the adult gut of five bilaterian species, to compare AP gene expression patterns and identify regional equivalences. We seek to define an interpretative lens, based on molecular data, through which the adult bilaterian gut may best be understood. We find such an interpretative lens in an AP scaffold that is overall organized in two compartments, and along which are arranged a specific set of five conserved expression modules which otherwise underlie highly divergent AP organization of shared functions. Our

investigation has implications beyond comparative studies of adult gut function and evolution, potentially informing equivalent frameworks through which to understand their embryonic/ontogenetic basis, historical and evolutionary reconstructions on the likely adult configuration of the through-gutted ancestor, and offers a potential avenue to assess the homology of the alternative adult through-guts found in non-bilaterians.

## Results

### Section 1: Adult guts in annelids, hemichordates, echinoderms, cephalochordates, and vertebrates

We considered five marine organisms across main branches of the bilaterian evolutionary tree. As a representative of the protostomes, the annelid worm *Platynereis dumerilii* (Dumeril's clam worm, "Pd"; [55]). Among deuterostomes, within the Ambulacraria, the echinoderm *Strongylocentrotus purpuratus* (purple sea urchin, "Sp") and the hemichordate *Ptychodera flava* (acorn worm, "Pf"). For the chordate branch, the cephalochordate *Branchiostoma floridae* (Florida lancelet, "Bf"), and the vertebrate *Amphiprion ocellaris* (common clownfish, "Ao"; [56]). We then dissected out the entire adult gastrointestinal tract (with associated digestive organs if applicable) from each of the five species under consideration (with the exception of Pf, see Methods), and further subdivided each gastrointestinal tube into subsegments taken to span the entire length of each tract, from anterior to posterior (Fig 1A). All segments from each species were processed for RNA sequencing. Detailed documentation about the dissection criteria, the dissection process, transcriptome extraction, and sequencing are provided in Materials and methods, S1 and S2 Figs.

Having thus obtained anteroposteriorly-resolved transcriptomic data of the gastrointestinal tract of five bilaterian species, we then proceeded to investigate the broadscale structure of gene expression across each, aiming to identify conserved principles of bilaterian gut organization. We first aimed to investigate whether gene expression similarities, within each species, would allow us to summarize segments into a smaller set of equivalent "domains," whether such domains would reflect the AP position of the segments themselves (i.e., would group contiguous segments or not), and whether such broader domains would reveal a conserved higher-order AP compartmentalization of gene expression across bilaterian guts (e.g., a tripartite one). We alternatively hypothesize that we could recover instead a continuous, gradual shift in transcriptomic similarity scores traversing the entire AP length of the gastrointestinal tract, a configuration that may or may not then further reveal itself to be conserved across through-guts.

By analyzing the Spearman's Rank Correlation coefficients between gut segments, calculated on the top 2000 most variant genes expressed in each gut, we do identify a conserved organization of gene expression along all five guts, and one with a mixed profile between the two scenarios mentioned above (Fig 1B). Specifically, we note that anteriormost gut segments (Fig 1B gray lines; Pd mouth to esophagus, Sp stomach-1 to stomach-4, Pf pharynx-1 to pharynx-3, Bf endostyle and gills, Ao esophagus to stomach) can consistently be summarized as a single well-defined molecular compartment, which excludes expected highly-dissimilar pre-oral segments, see the case in Pf. Concomitantly, we observe that segments posterior to this first domain, up to the terminal most end of the gut, define a second compartment of graded AP similarity where transcriptional similarity decays with distance (Fig 1B light-blue lines; Pd transition-2 to hindgut, Sp intestine-1 to intestine-3, Pf hepatic-1 to intestine-3, Bf liver to hindgut, Ao pyloric caeca to posterior intestine). In other words, we consistently identify two main gut compartments in all five through-guts, which we describe here as an anterior "block" compartment and a posterior "gradient" compartment. In contrast to the maybe more conventional tripartite understanding of bilaterian guts, often also applied to the adult configuration, we only see such an organization in few of the species considered (Pd, Sp; whose grouping would be polyphyletic), and as a subregion at the posterior of the gradient compartment.

Principal Component Analysis (Fig 1C) does not resolve clearer-cut groupings (except for Sp). Across the top principal components, segments distribute instead into distinctive arch-shaped arrangements, which resolve the actual AP anatomical position of the segment across the gut tube. Critically, AP position would seem to be the explanatory factor underlying the main transcriptional variability within bilaterian guts, supporting AP ordination of gene expression as a shared feature

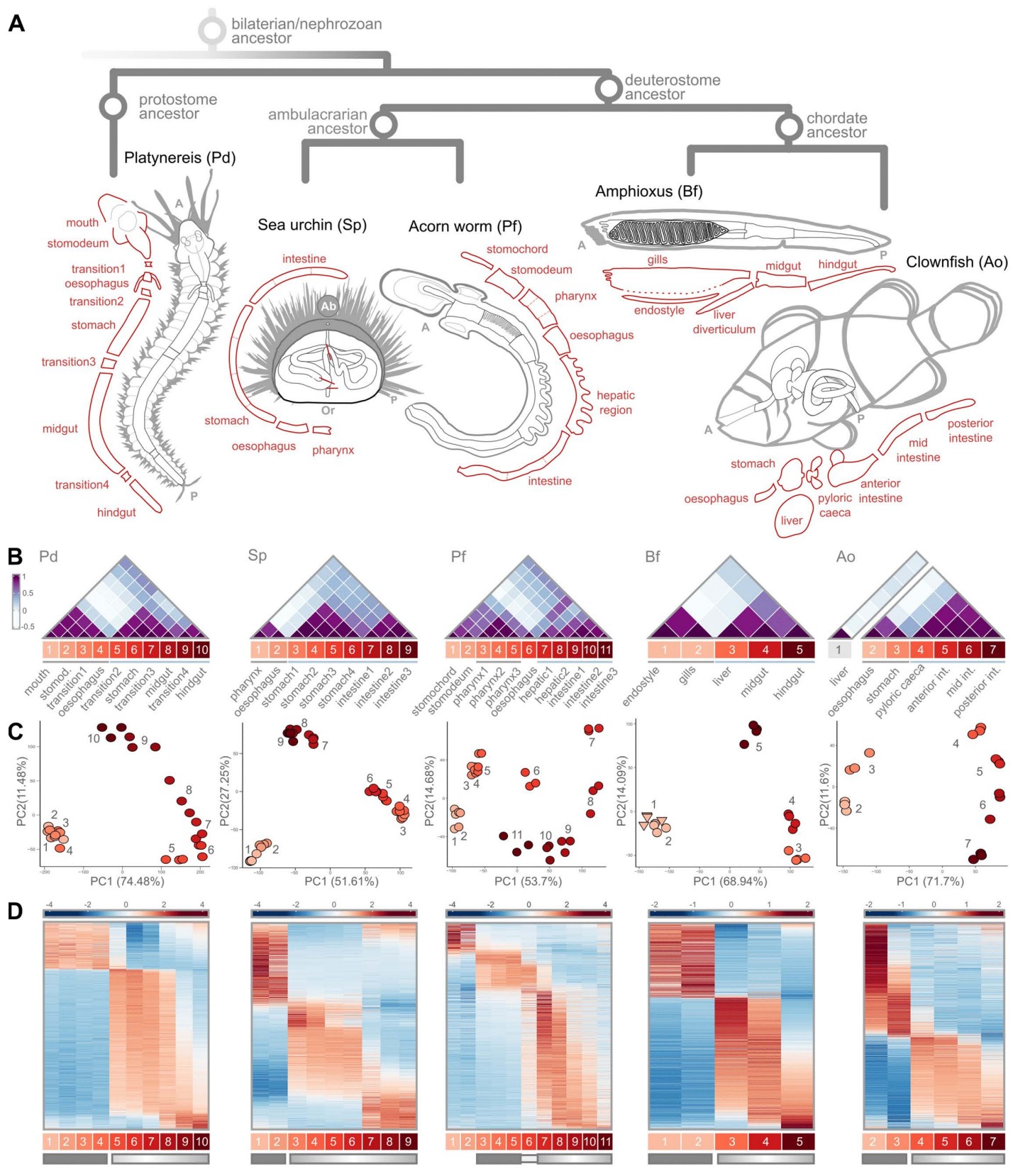

**Fig 1. Adult guts in annelids, hemichordates, echinoderms, cephalochordates, and vertebrates. A)** Schematic phylogeny of the 5 bilaterian species compared in this study, and of their gastrointestinal anatomy. In red, schematic illustration of the gut segments collected, arranged along the gut AP axis. A, anterior; P, posterior; Or, oral; Ab, aboral. **B)** For each species, heatmap of the Spearman's Rank Correlation coefficients between all pairs of gut segments (white, low; purple, high). Segments are numbered sequentially and color-coded according to AP position (where applicable). Gray line: segments of the first compartment. Light blue line: segments of the second compartment. **C)** For each species, distribution of gut segments (all replicates) across the two main Principal Components (PCs; PC1, PC2). Segments are color-coded and numbered by increasing AP position, as in B. The clownfish liver segment is omitted because it dominated the variance of the combined dataset. **D)** For each species, heatmap showing the expression pattern (z-scores) of the top 2000 most-variant genes along the gut (columns: AP-ordered gut segments). Genes (rows) are ordered according to Rank-2 Elliptical seriation (R2E). Dark rectangle: segments classified as "block compartment"; Graded rectangle: segments classified as "gradient compartment" (see main text). Here, again the expression data for the clownfish liver segment is not included as the organ is not part of the adult AP structure. The data underlying this Figure can be found in https://doi.org/10.5281/zenodo.17746910.

of bilaterian guts. In all species considered, the first axis of transcriptional variation (PC1) would correspond to differences between anterior segments and segments just posterior to them (i.e., anterior segments of the gradient compartment), while the second axis of variation (PC2) would seem to reflect gene-expression patterns characteristic of terminal segments. Interestingly, the most dissimilar regions of the through-gut would not be the most distant ones (mouth + esophagus versus the hindgut/anus), but rather the anterior versus middle regions.

We stress, however, that arch/horseshoe-shaped patterns in PCA can result from well-known properties of autocorrelated data [57], and can lead to erroneous interpretations of sample similarity and of the effective drivers of sample differences [58–62]. We note that we could not rule out nor confirm that the above segment (dis)similarity inferences are artefactual. For this reason, and because the degree of similarity between bilaterian through-gut termini has particular importance in gut EvoDevo, we preferred to root our assessment of segment affinities on approaches that do not rely on dimensionality reduction throughout the rest of this work.

We therefore proceeded to visualize the actual patterns of the genes expressed across through-gut segments (Fig 1D). We use heatmaps (which are indeed a dimensionality-reduction-free approach), and combine them with a seriation criterion able to highlight long-range gradients in expression pattern (Rank-2 Elliptical seriation, "R2E"; [63]; see dedicated section in the Materials and methods). We expect such long-range gradients in expression patterns from our sample-correlation analysis, PCA, and the increasing literature on gene expression organization across adult guts [64–66]. We observed that the main organization of gene expression along all guts indeed consistently highlights a first set of genes marking anterior segments and a second group of genes with much more pronounced graded distribution spanning the segments of the second block.

Since our transcriptomic data can be directly linked to the actual anatomical location of the segments, we clearly notice that the boundary defining the separation between these two conserved through-gut domains invariably corresponds to an actual anatomical boundary in the form of a sphincter. This sphincter—which we would suggest to be a key reference transition point between two potentially comparable regions of bilaterian through-gut architecture—is not considered to be an equivalent structure across bilaterians ("pyloric sphincter" in vertebrates, and the transition between esophagus and stomach in sea urchin/invertebrates). In this new interpretative key, we would like to refer to it as the "transition sphincter".

Studies of the *Drosophila* through-gut, where meticulous characterization of the correspondences between anatomical sphincters and gene expression boundaries is available [64], similarly identify the anterior "cardiac" sphincter as the anatomical site of the majority of gene expression boundaries, i.e., the site of highest gene expression discontinuity. In conclusion, we find bilaterian adult guts to be best summarized as "anterior-block, posterior gradient" systems, with the transition sphincter in between.

### Section 2: Hox and ParaHox patterning systems in adult guts in bilaterian lineages

Hox genes are the most likely candidate for an AP reference system along the bilaterian gut. Though the investigation of their axial expression patterns has extensively focused on neuroectodermal and mesodermal tissues, it has long been

put forward that the expression pattern in these tissues may have been just a later recruitment of an original patterning system of the gut tube [67]. Moreover, Hox genes are notably amongst the few embryonic gut patterning genes whose expression is well known to persist in the adult gut [68,69] to the point that detailed maps of their expression patterns for key vertebrate models are widely available [69].

Based on the most recent genomic annotations, we first determined a conservative reconstruction of the full Hox complement of the common clownfish (see Materials and methods; 47 Hox genes, 38 of which expressed in the gut). We further verified previously published reconstructions of the Hox complement in the other four species under consideration [70–73] (Fig 2A). We are then able to describe the complete map of anteroposterior expression of Hox genes across the entire adult gut of the five bilaterian species considered (Fig 2B).

We find that Hox genes are expressed along the gut of all bilaterian species considered, and we recover an overall collinear AP pattern of gene expression in all species, even in species whose Hox genes are not genomically collinear such as Sp (Fig 2B; [70]). Anteriormost Hox genes are mostly overlappingly expressed as a single block within anterior segments; central and posterior Hox genes are predominantly expressed in segments posterior to those, in a nested, graded fashion that defines increasingly posterior segments and coincide with the graded compartment discussed in Section 1 (Fig 2B). Critically, we therefore find a correspondence between the block-gradient expression pattern of Hox genes and the overall block-gradient organization of gene expression along the gut (Section 1). We are able to identify other recurring features of the Hox patterning system across bilaterian guts: i) segments that are not part of the actual linear AP sequence of the gut tube (bifurcations and separated organs) tend to be excluded by the Hox patterning system (see Bf hepatic cecum, Ao liver), and ii) the anteriormost segments of the gradient compartment consistently show lowest expression of Hox genes, and lowest number of Hox genes expressed (see Pd t2 and stomach, Sp stomach1/2/3, Pf hepatic1, Bf midgut, Ao pyloric caeca and Anterior intestine). In this sense, the expression of middle Hox genes is often shifted away from the posteriormost boundary of expression of anterior Hox genes, leaving intervening segments Hox-poor (one exception is Pf, likely due to the inclusion of the ectodermal layers that obscure the analysis result). We finally note that Hox genes also allow to define an anteriormost terminus of the gut AP system, with pre-oral and oral structures generally not showing anterior Hox gene expression (see Pd mouth and stomodeum, Sp pharynx, and Pf pre-oral segments).

This same pattern (anteriormost Hox genes overlapping in the anterior gut, posteriormost genes grading the posterior, and relative paucity of Hox genes expressed in the intervening segments) is indeed the pattern recovered for the embryonic gut of the mouse [74–76]. We here recover it as a conserved signature of adult bilaterian through-guts, therefore locating the gut Hox code as a likely ancestral feature of the bilaterian ancestor, specifically in the deployment pattern highlighted above.

We then investigated the pattern of expression of ParaHox genes *gsx*, *pdx*, *cdx* (Fig 2C). We confirm the general exclusion of *gsx* as part of the bilaterian gut patterning system. In none of the adult guts here considered, does *gsx* appear to carry anterior information. We confirm instead that the classic ParaHox gut patterning system of sequential *pdx* and *cdx* expression is a distinctive feature of adult bilaterian guts, and we note that these two ParaHox genes are specifically deployed to pattern the gradient compartment and not the block compartment. Critically, with *pdx*, the ParaHox system fills in the Hox-poor, intermediate segments, in fact most often allowing to determine the anteriormost boundary of the gradient compartment. Gut tube bifurcations and separated organs are excluded from the ParaHox system, just as they are from the Hox system (see again Bf hepatic cecum, Ao liver).

In summary, we find that adult bilaterian guts share a common complementary Hox and Para-Hox AP organization system, deployed on a bipartite basis just like the overall gene expression patterns, and that only includes segments that are part of the actual linear AP sequence of the gut. With an anterior boundary that does not necessarily match the anteriormost segments of the gastrointestinal tube, bilaterian guts share a Hox and Para-Hox AP coordinate system of i) segments characterized by block expression of anterior Hox genes, and no ParaHox, ii) segments characterized by *pdx* expression with low levels of Hox expression, iii) segments characterized by graded expression of posterior ParaHox (*cdx*)

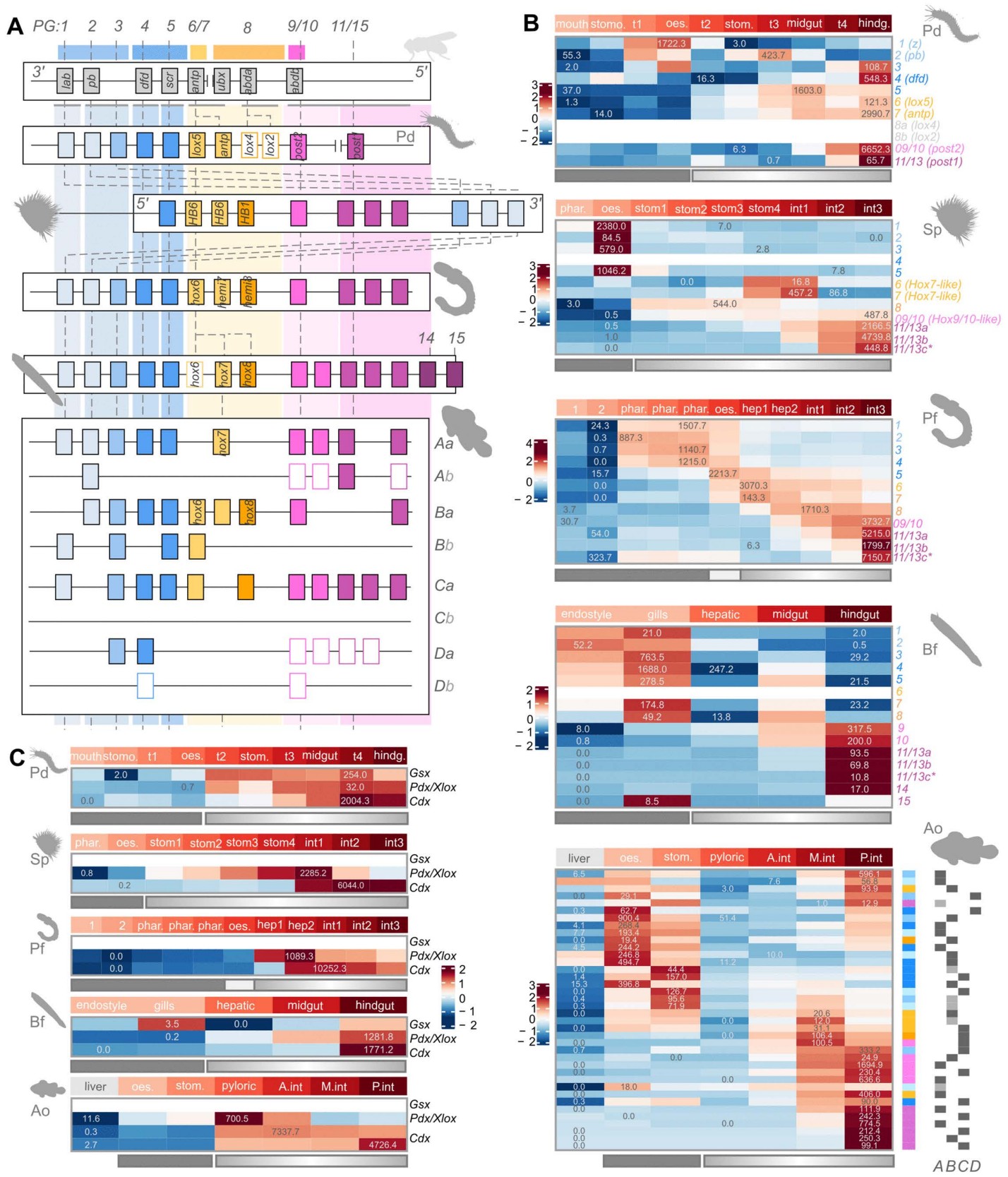

**Fig 2. Hox and ParaHox patterning systems in adult guts in bilaterian lineages. A)** Reconstruction of the Hox complement of the 5 species under study, in reference to the fruit fly model (*Drosophila melanogaster*) reference. Hox genes are color-coded by general AP position from blue to yellow/orange, to pink/purple. Hollowed rectangles indicated absence of expression of the corresponding Hox gene in the gut dataset. PG, paralogous group. **B)** For each species, heatmap showing the expression pattern (z-scores) of all expressed Hox genes along the gut AP axis. Genes (rows) are ordered according to Hox paralogue number, except in clownfish where they are ordered according to R2E. **C)** For each species, heatmap showing the expression pattern (z-scores) of all expressed ParaHox genes along the gut AP axis. Overlayed numbers indicate the transcripts per million corresponding to the underlying z-score (highest and lowest). Solid and graded rectangles indicate segments previously assigned to block and gradient gut compartments, as defined in previous sections. The data underlying this Figure can be found in https://doi.org/10.5281/zenodo.17746910.

and posterior Hox genes. The anterior Hox segment corresponds to the posterior region of the anterior block compartment, while the *pdx* segment and the *cdx*/Hox segment coincide with the posterior gradient compartment.

## Section 3: Larval AP patterning genes display regionalized expression pattern along adult guts in bilaterian invertebrates

Having recovered the persistent expression of Hox and ParaHox embryonic patterning genes along the adult gut of all the species considered, we investigated the extent to which this might be a conserved feature of other genes classically associated with gut tube AP patterning. Traditionally, these genes have been studied (and defined) based on embryonic or larval stages. Given the absence of established sets of conserved adult gastrointestinal AP patterning transcription factors (TFs), we resort to these classical embryonic signatures, hypothesizing that they might also be recovered in adult guts just as Hox and ParaHox genes.

We therefore refer to the set of conserved TFs defined in Annunziata and colleagues [37], shown to pattern the AP axis of the developing gut tube of early sea urchin larvae and as such representing a set of evolutionary conserved gut patterning markers shared with vertebrate embryos [37] and bilaterian embryos more generally [38,77]. We show that the expression of all of these early patterning markers (including, here again, Hox and ParaHox genes) is conserved along the *Strongylocentrotus purpuratus* adult gut, virtually unchanged (Fig 3A). Such conservation has also most recently been shown in the post-metamorphic (juvenile) gut of a close species, *Paracentrotus lividus* [78]. We find this conservation extraordinary as it suggests the long-term permanence or re-establishment of a "larval" AP gut signature long into adulthood. Furthermore, such signature is manifestly retained/redeployed even across a drastic ontogenic restructuring of the gut tube itself, as is the case of what happens specifically to the sea urchin gut tube during metamorphosis [79].

Crucially, we further show that these same markers display similar AP expression patterns in the adult gut of our other ambulacrarian species *Ptychodera flava* (Pf, Fig 3B), as well as the adult gut of the two chordate species, despite an increasing loss of resolving power in this branch. We show that this TF signature is extremely well conserved in the adult gut of the protostome *Platynereis dumerilii*, hinting at a deep conservation of this signature (and possibly in turn, of its underlying early embryonic/larval patterning basis) at least up to the bilaterian stem.

In addition to such remarkable conservation, we note that where there are differences in pattern across species, these often involve anterior gut markers (in Sp) shifting to the posterior or showing expression at anterior *and* posterior in other species. Conversely, some posterior gut markers (in Sp) shift to the anterior, or show anterior *and* posterior expression in other species. In other words, a subset of anteriormost or posteriormost gut markers, which would appear as having poor AP conservation, may be best understood at a more holistic pan-bilaterian level as "terminal" markers (see e.g. *foxD*, *foxP*, *foxI*, Fig 3B), and be conserved in this role across life stages and species. "Terminal" markers, in bilaterian through-guts, would therefore be markers that in some species are "anterior" markers, in some species are "posterior" markers, and in others more are "anterior and posterior" markers. At the pan-bilaterian level, they are "markers of the termini (anterior gut and hindgut)", across species or across life-stages.

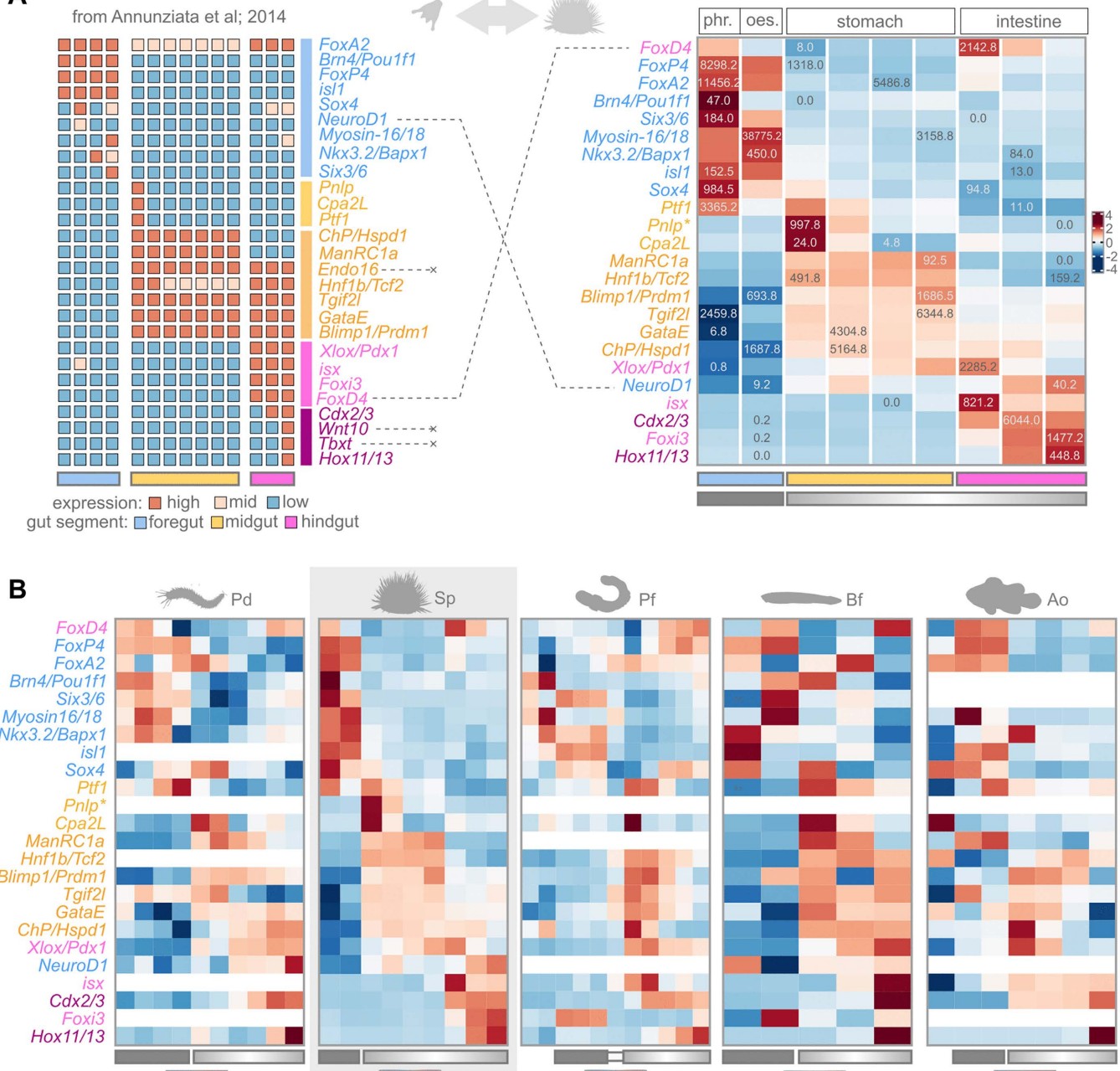

**Fig 3. Larval AP patterning genes display regionalized expression pattern along adult guts in bilaterian invertebrates. A) Left**: qualitative "heatmap" showing the expression pattern (red, high; blue, low) of conserved gut patterning markers along the sea urchin larval gut; as per Annunziata and colleagues [80]. Redrawn from the same publication. Blue rectangle, foregut, yellow rectangle, midgut; pink rectangle, hindgut. **Right**: heatmap showing the expression pattern (z-scores) of the same marker genes along the sea urchin adult gut AP axis (this study). Genes (rows) are ordered according to R2E seriation. Blue, yellow, pink rectangles, segments inferred to correspond to the larval foregut, midgut, hindgut, respectively. Dashed lines link genes with shifts in AP expression patterns across datasets. Truncated links indicate genes that are not expressed in the adult dataset. **B)** For each species, heatmap showing the expression pattern (z-scores) of all expressed gut patterning genes along the gut AP axis. Genes (rows) are ordered according to the adult sea urchin reference sequence (highlighted in gray). Overlayed numbers indicate the transcripts per million corresponding to the underlying z-score (highest and lowest). Solid and graded rectangles under the heatmaps indicate segments previously assigned to block and gradient gut compartments, as defined in previous sections. The data underlying this Figure can be found in https://doi.org/10.5281/zenodo.17746910.

In conclusion, we show a remarkable conservation of the embryonic/larval gut plan into the adult: adult bilaterian guts maintain or redeploy "embryonic" and "larval" markers and do so in a way that parallels known embryonic or larval AP patterns of expression. To some extent, these results expand, at least for the gut tube, the applicability of these markers and question conceptions of embryonic patterning systems as transient or impermanent. Given that the gut tube of some of the species we consider above undergoes dramatic restructuring during metamorphosis, these results also bring forward new questions about the possible mechanisms and evolutionary constraints that would bring about the maintenance or redeployment of AP expression patterns across an impermanent patterning substrate. Regardless, we show that the AP pattern of expression of these markers (with the modification we document above and with frequent terminal shifts) is strongly conserved across bilaterians, and in fact even in the annelid worm *Platynereis*, hinting at a possible feature of the ancestral adult gut tube.

## Section 4: Unbiased analyses of regionalized gene expression along the adult gut in bilaterian lineages

In an attempt to identify an adult gut AP signature conserved across bilaterians, we decided to map the expression pattern of *all* TFs along the AP axis of our adult bilaterian guts and see if we could identify a shared organization across species. This strategy uniquely allows us to approach this question in an unbiased, systematic, and comprehensive way, within the limits of our comparative approach and gene orthology attributions.

We identified all TFs within each species' proteome by a protein domain scan referencing the AnimalTFDB database (see Materials and methods). The proteins we retrieved (proteins with TF domains, which we here take as "TFs") were then summarized at the orthogroup level to allow between-species comparisons. To avoid discarding orthogroups with many-to-one correspondences, a "best ancestral orthologue" representative was selected among a species' paralogues based on sequence similarity scores, as in recent literature ([81]; see Materials and methods). Of these TF orthogroups, 356 were shared across all five bilaterian species, in line with the minimal TFs content estimated for the Protostome-Deuterostome Ancestor (216; [82]). 277 out of these 356 TFs were expressed in all five gut datasets (Fig 4A).

We observe that R2E seriation of gut segment TF expression within each species orders them based on their actual anatomical location (S3A Fig), whereas R2E seriation of gut segments across species does not (S3B Fig). The same overall non-equivalence is seen in the grouping of samples across PCA space (S4A Fig). Specifically, it appears that when considering all conserved expressed TFs across species, Pd hindgut segments group with bilaterian foreguts, and intestinal segments (especially of the two chordate species) do not necessarily cluster close to their expected AP equivalents in other species (S3 and S4 Figs). In other words, beyond a pan-bilaterian subdivision that allows to classify segments in "block compartments" (with Pd hindgut; *gli2+*, *znf521+*, *zeb2+*, *six1+*), and "gradient compartments" (*nr2a1*/*hnf4a+*, *srebf+*, *zfp36l+*, *pdx1+*, *gata4+*, *Hox PG09+*), we recover an overall high divergence (non-equivalence) of the AP organization of shared TFs along adult guts. At the same time, we do observe a minority subset of TFs that do appear to define segments at equivalent (anatomical) AP positions (S3 Fig, panel B). Orthogonal network-based approaches able to consider all TF and TF paralogues (SAMap, [83]) recover finer AP matches on a pair-by-pair basis, yet similarly highlight an overall fragility of pan-bilaterian AP correspondences (S5 Fig, upper triangle).

Observing that only a minority of TFs across bilaterian guts define the coordinates of the anatomical AP axis, we decided to isolate such TFs by modeling gene expression profiles with generalized additive models and identifying genes whose expression pattern is most associated with AP position (Tradeseq; [84]). By reclustering all segments based on this refined subset of 85 TFs, segments do cluster by relative AP position, with overall much stronger cross-species concordance between anterior, middle, and posterior segments of the gradient compartment (Fig 4B). Overall, we are therefore able to deconstruct the common bilaterian AP axis into 5 (non-exhaustive) main gene sets ("modules"; Fig 4B and 4C).

1. An "anterior (only)" module (module 1) characterized by known hox genes PG01 and PG02, and which includes *nkx2.3*, *smad3*, and *camta1*. This module groups all the segments anterior to the "transition sphincter," and matches the

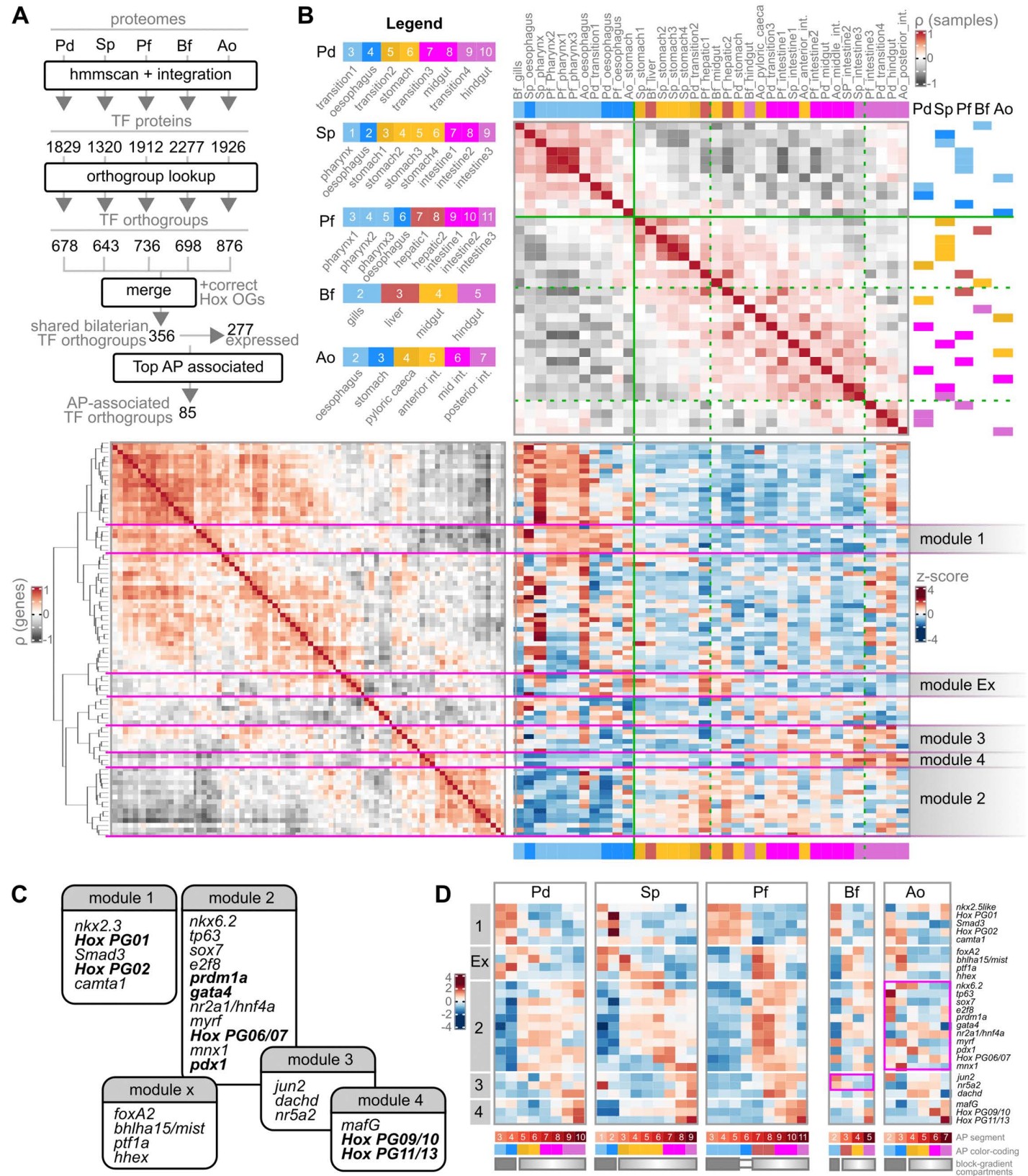

**Fig 4. Unbiased analyses of regionalized gene expression along the adult gut in bilaterian lineages. A)** Summary of the steps deployed to derive the final set of conserved, gut-expressed transcription factors (TFs) used for cross-species comparison. **B)** Generalized Association Plot summarizing

the expression data of the 85 AP-associated TF orthogroups across bilaterian gut segments. **Top**: heatmap of the Spearman's Rank Correlation coefficients between all segments, regardless of species. Segments are ordered according to R2E seriation. **Bottom Left**: heatmap of the Spearman's Rank Correlation coefficients between all 85 AP-associated TF orthogroups considered, ordered according to R2E seriation-guided hierarchical clustering. **Bottom Right**: heatmap showing the expression pattern (z-scores) of all TFs considered based on the segment (columns) and gene (rows) order defined above. Segments are color-coded by approximate equivalent AP position as in the Legend provided. Solid and dashed green lines indicate qualitative assignment of major and minor groupings of gut segments based on correlation patterns and known anatomical position. Magenta lines identify gene modules. **C)** Summary of the 5 main modules identified, in their general AP order, with representative TF markers indicated. In bold, established gut patterning markers across the literature. **D)** For each species, heatmap showing the expression pattern (z-scores) of all representative TFs from each of the identified modules along the gut AP axis. Magenta highlights: modules that appear not to match the expected expression pattern in a given species. Solid and graded rectangles under the heatmaps indicate segments previously assigned to block and gradient gut compartments, as defined in previous sections. The data underlying this Figure can be found in https://doi.org/10.5281/zenodo.17746910.

domain identified as the 1st compartment ("block compartment") of the shared bilaterian bipartite organization of gene expression.

2. an "intestinal/digestive" module (module 2) characterized by known ParaHox gene *pdx1*/*xlox*, *gata4*, *hnf4*, *prdm1a*, and also several novel potential markers (*sox7*, *myrf*, and *e2f8*).

3. an "intermediate module" (module 3) generally expressed in more posterior segments, and notably characterized by *nr5a2* and *jun2* expression, shifted to the gills in Bf.

4. a "posterior" module (module 4) characterized by an extremely limited set of members, and which are indeed just *Hox PG08/09* and *Hox PG11/13*, as well as *mafG* (which is, however, already not conserved in Bf). This module groups all the posteriormost segments of all species. Note that *cdx* could not be considered in this unbiased analysis (see Materials and methods "Filtering and standard counts processing") but would also be another marker of this module as shown in previous figures (e.g., Fig 2).

5. We also note a fifth module that we here call module "Ex" (for "Exocrine") due to its grouping of notable gut patterning TFs *ptf1*, *hhex*, and *bhlha15*/*mist* associated—in vertebrates—with pancreatic tip acinar cells [85]. This would represent a module clearly of key interest, overall conserved AP position, but variable position with respect to bilaterian gut compartments and the modules that characterize them. In Pd and Ao, it is detected in posterior segments of the block compartment (i.e., esophageal segments, including the vertebrate stomach; i.e., overlapping module 1 segments), in the other species it is detected at the anteriormost segments of the gradient compartment, i.e., overlapping with module 2 segments.

Comparing the gene expression pattern of representatives of these modules confirms their validity as conserved bilaterian AP markers, while also identifying again significant elaboration within the chordates (Fig 4D). We critically note that module 2 is highly fragmented in the gut tube of clownfish (Ao, magenta outline) and is in big part represented instead by the esophagus ("anteriorisation"). Similarly, we note a cephalochordate-specific "loss" of module 3 with a shift to the gills (Bf, magenta outline).

Overall, we here define a conserved set of TFs that mark AP positions which we would consider to be homologous in adult bilaterian guts. Remarkably, the signature we identify, and which we summarize here in 5 main modules, appears to be conserved across hundreds of millions of years of evolution and despite the divergence of overall TF expression patterns, through-gut anatomies, and dietary specialization.

We stress that this list of TFs is a conservative list of markers, not to be interpreted as precluding the existence of additional TFs with a conserved AP expression in these five bilaterian species. Specifically, such expression patterns may be conserved in paralogues that were here discarded when resolving many-to-one correspondences. Though such a loss of potentially informative TFs is inherent to comparative transcriptomic approaches [81,83,86,87], we note that it is here

additionally complicated by the underlying non-established equivalence between gut segments. Critically, this prevents more refined orthologue selection criteria that also take into account the conservation of tissue expression pattern.

We also continue to recover, especially at this cross-species level, an overall high transcriptional similarity between segments at the two ends of the through-gut. This pattern echoes our previous recovery of anterior or posterior TFs in some species as markers of the opposite terminus (or of both) in other species, and ontogenetic inter-termini shifts of gene expression at least in sea urchin. At the pan-bilaterian level, the shared expression of terminal Hox genes may well be one of the few conserved features of bilaterian through-guts.

## Section 5: Common functions of bilaterian through-guts

We repeat the analysis on non-TF genes, selecting the top 2,500 most variant genes out of a total of 5,562 non-TF genes expressed in the through-gut of all 5 species. Critically, the main interpretation of gut segments based on these other (functional/structural) genes, at the cross-species level, is one where anterior functions/structures are distinct from hepatic and anterior-intestine segments, but where posterior segments have highly variable affinities across species (Fig 5A and 5B). In this regard, the hindgut of the protostome Pd is not an entity distinct from anterior through-gut segments (if only in space). This is also the case for Pf hindgut segments, a deuterostome, indicating that observed terminal equivalences may not necessarily directly relate to the embryonic origin of these segments. Cross-species comparison based on SAMap highlights cross- or U-shaped alignments centered on the hepatic/anterior intestinal segments (S5 Fig, lower

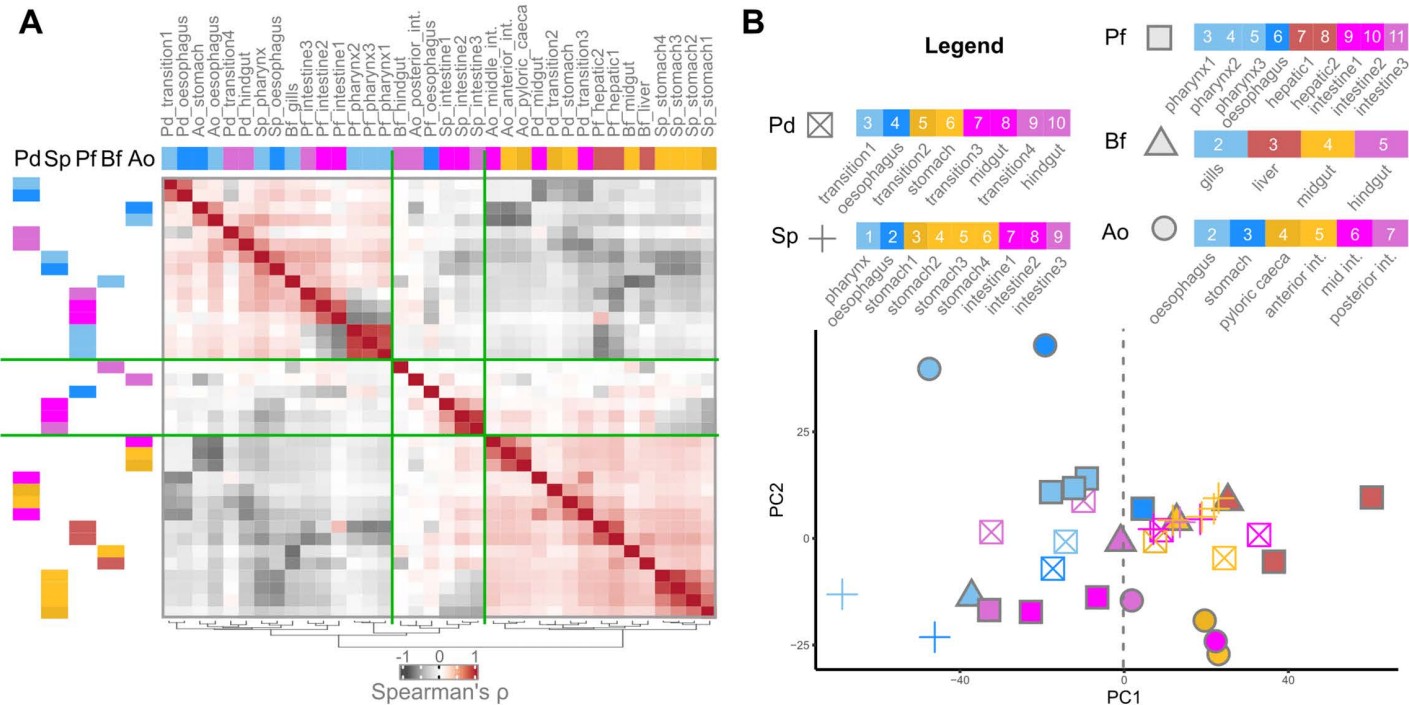

**Fig 5. Analysis of adult bilaterian gut segments based on non-TF gene expression. A)** Heatmap of the Spearman's Rank Correlation coefficients between all gut segments, regardless of species, based on the full set of conserved, gut-expressed non-transcription factors. Segments are ordered according to R2E seriation. Green lines indicate qualitative assignment of major and minor groupings of gut segments based on correlation patterns and known anatomical position. **B)** Distribution of all gut segments (all species) across the two main Principal Components (PCs; PC1, PC2) based on the same set of genes. Segments are color-coded by approximate equivalent AP position as in legend. Dashed vertical line indicates main separation between clusters (PC1 = 0). The data underlying this Figure can be found in https://doi.org/10.5281/zenodo.17746910.

triangle) indicating an even lower AP concordance between segments of different species than that instead highlighted by TF expression. Here again, through-gut termini (anterior *and* posterior) would in many cases be considered equivalent matches across species.

Not surprisingly, we find that bilaterian through-guts can be summarized as sites of protein synthesis and secretion (translation, protein transport, intracellular protein transport), lipid transport, and fatty acid, carbohydrate, and amino acid metabolism (S6A Fig). We find that many of these functions are highly localized within the AP length of the gut of each species, and localized to comparable AP positions across species (S6B Fig). We note, for example, that, in all bilaterians considered, translation functions are consistently localized to the first segments of the gradient compartment, likely reflecting an AP constraint in the site of digestive enzyme synthesis/secretion. In Ao, this function is correspondingly mainly localized to the liver, which releases its enzymes in these same segments. Fatty acid metabolism is also highlighted as a conserved function enriched in the anterior portion of the gradient compartment in all bilaterian guts considered. At the same time, significant re-elaboration (shifts) and losses of functions can be seen even just within the five species sampled, reflecting not only the overall poor conservation of AP localization of gut functions but also the remarkable versatility in how these functions are spatially deployed.

### Section 6: Characterization of bilaterian gut modules

We turn to a more in-depth characterization of the modules we identified based on TF expression patterns, in an attempt not only to build a more complete picture on the markers that define them, but also to identify functions that may be conserved in their association with homologous segments rather than in their relative AP position. Critically, our module-based interpretative framework allows us to impute shared segment identities across species, and therefore to apply *supervised* dimensionality reduction approaches such as Partial Least Squares-Discriminant Analysis (sPLS-DA, [88]; see dedicated section in the Materials and methods) to identify their discriminating markers and/or expand the signatures beyond TFs alone.

**Module 1/"cilia".** Module 1 corresponds to segments that are consistently located at the anterior of the gut tube of all bilaterian species considered, and that in our bipartite interpretative lens represents the anterior, Hox-positive, "block" compartment. We therefore impute a shared identity to all of these segments and apply sPLS-DA to find discriminating markers (Fig 6). Among the highest scoring markers we note, in addition to anterior Hox genes and more SMAD signaling components, a dominant representation of Hedgehog signaling components (*patched 2*, *smoothened*, but also *GLI family zinc finger 2a*) and primary cilia or cilia/microtubule-related genes (gamma tubulin complex associated protein genes, *centrosomal protein 83*, *testis-associated actin remodeling kinase*). Accordingly, while the functional interpretation of this anteriormost module as a ciliated, filtering module would be intuitive, our analysis highlights hedgehog signaling (and therefore primary cilia) as the most distinctive feature of this module. Given the fundamental role played by hedgehog signaling during embryonic foregut development [89], it is remarkable that we recover once again embryonic signatures (now in the form of signaling pathways) in adult guts, and that we do so in the form of pan-bilaterian conserved, AP-conserved patterns. We also note that other lesser-known markers of this module, whose pan-bilaterian conservation would highlight as of particular relevance, such as *Sin3a*, have indeed been linked to profound, targeted developmental anomalies within the foregut [90].

**Module 2/"digestive".** In our reconstructed pan-bilaterian AP signature Module 2 defines a domain that is at the anterior of the gradient compartment of bilaterian guts (the invertebrate "stomach"), and which in vertebrates seems to be fragmented across the esophagus, stomach, and anterior intestine (Fig 4D). Expanding the signature of Module 2, by performing sPLS-DA with clownfish esophagus rather than anterior intestinal segments, identifies several factors also shared with the vertebrate stomach and the liver, mostly related to protein synthesis, but also amino acid metabolism, and ether lipid synthesis. These would represent strong candidates for true module-associated functions, yet the fragmentation

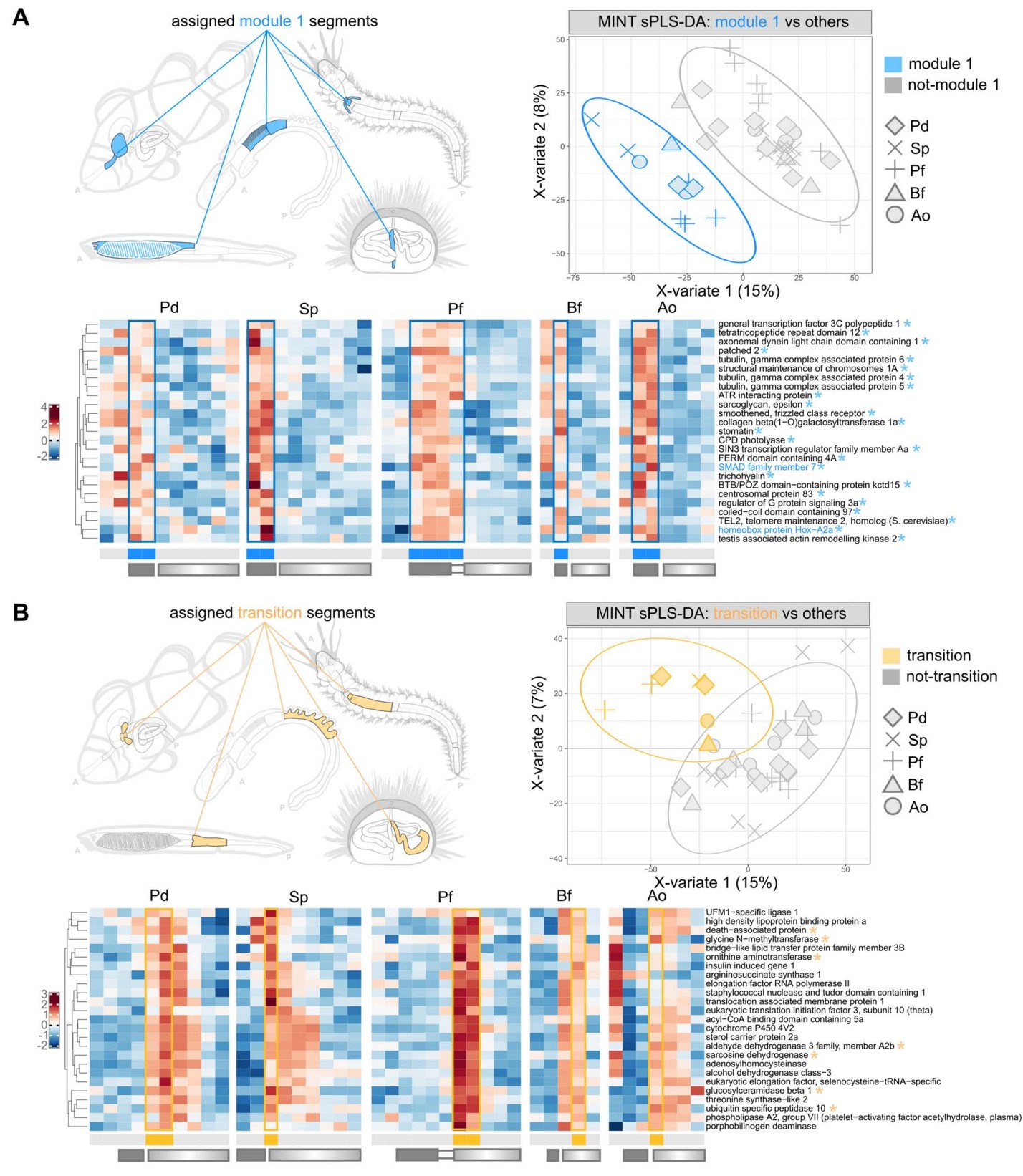

**Fig 6. Module 1 and transition sPLS-DA. A**) sPLS-DA of bilaterian gut segments marked by Module 1 TFs. **Left:** illustration of the segments to be discriminated (Module 1 segments). **Right:** separation of segments according to sPLS-DA discriminant genes. **Bottom**: For each species, heatmap showing the expression pattern (z-scores) of the top discriminant genes identified through sPLS-DA. Genes (rows) are clustered according to hierarchical clustering. Segments (columns) are ordered according to AP position. Solid and graded rectangles under the heatmaps indicate segments previously assigned to block and gradient gut compartments, as defined in previous sections. Gene names followed by an asterisk constitute the optimal sufficient discriminant set. Gene names in color are TFs. **B**) sPLS-DA of bilaterian gut segments marked by equivalent anatomical positions (and Module 2 TFs in all species except Ao). **Left:** illustration of the segments to be discriminated ("transition" segments, all anteriormost segments of the gradient compartment; yellow highlight). **Right:** separation of segments according to sPLS-DA discriminant genes. **Bottom:** For each species, heatmap showing the expression pattern (z-scores) of the top discriminant genes identified through sPLS-DA. Genes (rows) are clustered according to hierarchical clustering. Segments (columns) are ordered according to AP position. Solid and graded rectangles under the heatmaps indicate segments previously assigned to block and gradient gut compartments, as defined in previous sections. Gene names followed by an asterisk constitute the optimal sufficient discriminant set. Gene names in color are TFs. The data underlying this Figure can be found in https://doi.org/10.5281/zenodo.17746910.

of this module across multiple segments in Ao may be likely confounding this analysis (which for this reason we relegate to S7 Fig).

We note instead that the vertebrate (Ao) anterior intestine, consistently appears more affine to more posterior invertebrate segments (Module 3; S5 Fig), and yet ends up abutting the same equivalent domain of other species' module 2 segments (i.e., abutting Module 1). We therefore wondered whether we could identify a molecular signature common to such an equivalent anatomical position and accordingly performed sPLS-DA by coercing vertebrate pyloric caeca into sharing an identity with invertebrate Module 2 segments. Notably, we find that a notable subset of these (positional) markers is in fact expressed in the clownfish liver, or in both the liver and the anterior intestine (Fig 6), thus reflecting the known ontogeny of this organ in vertebrates. Markers of this position include many mitochondrial and non-mitochondrial genes generally consistent with a conserved role in the metabolism of sterols and amino acids (*insulin induced gene 1, sterol carrier protein 2a, ornithine aminotransferase, sarcosine dehydrogenase*), including what would be traditionally classified as stereotypical "liver" function markers (*glycine N-methyltransferase*), and consistent with the previously described enrichment of translation and fatty acid metabolism at this location.

**Module 3/"sensor".** We analyze Module 3, which overall characterized through-gut segments of the "gradient compartment" generally more posterior than those characterized by Module 2, in all species (Fig 4B). Very surprisingly, the module is absent from the amphioxus gut, and localizes instead to the gill. We use sPLS-DA to identify common discriminant markers of these segments, which we consider homologous (therefore, including Bf gills), and identify even more genes with likely conserved roles as metabolic integrators and regulators (Fig 7). These include the glucose sensor *mlxip*/*MondoA*, SREBF pathway regulator in Golgi 1 *Spring1*, a progestin and adipoQ receptor family member, but also *ketohexokinase*, as well as numerous other TFs (*foxO3, ncoa2, pknox1.1*). All are associated with the segments with *Hox6/7* expression.

We attempted to identify a more precise function (or clear vertebrate reference) to this cluster, and yet literature on these markers (first and foremost, *Nr5a2*) consistently recover associations with both liver and pancreatic development, and adult endocrine functions [91–93]. We also note that the conserved localization of this module posterior to the *pdx* boundary and within the anterior of the *cdx* domain, the endocrine and pancreatic association of *Nr5a2* in vertebrates, the overall mixed metabolic-endocrine signature of other conserved markers we recover, and the extremely restricted expression of this module in the adult sea urchin, would suggest a most likely correspondent in a recently-described subpopulation of "endocrine pancreas" cells known to localize posteriorly to the pyloric sphincter in the larval sea urchin [49]. We also note that *Nr5a2* has been critically identified as a key regional marker along the *Ciona intestinalis* juvenile gut, again with localized expression in the *cdx*+ region posterior to the *pdx* boundary [94–96]. Remarkably, this posterior expression (but not a more anterior one) does not appear to be constitutive in the gut [94,96], hinting again at a "responsive" function of the posterior regions of the gut.

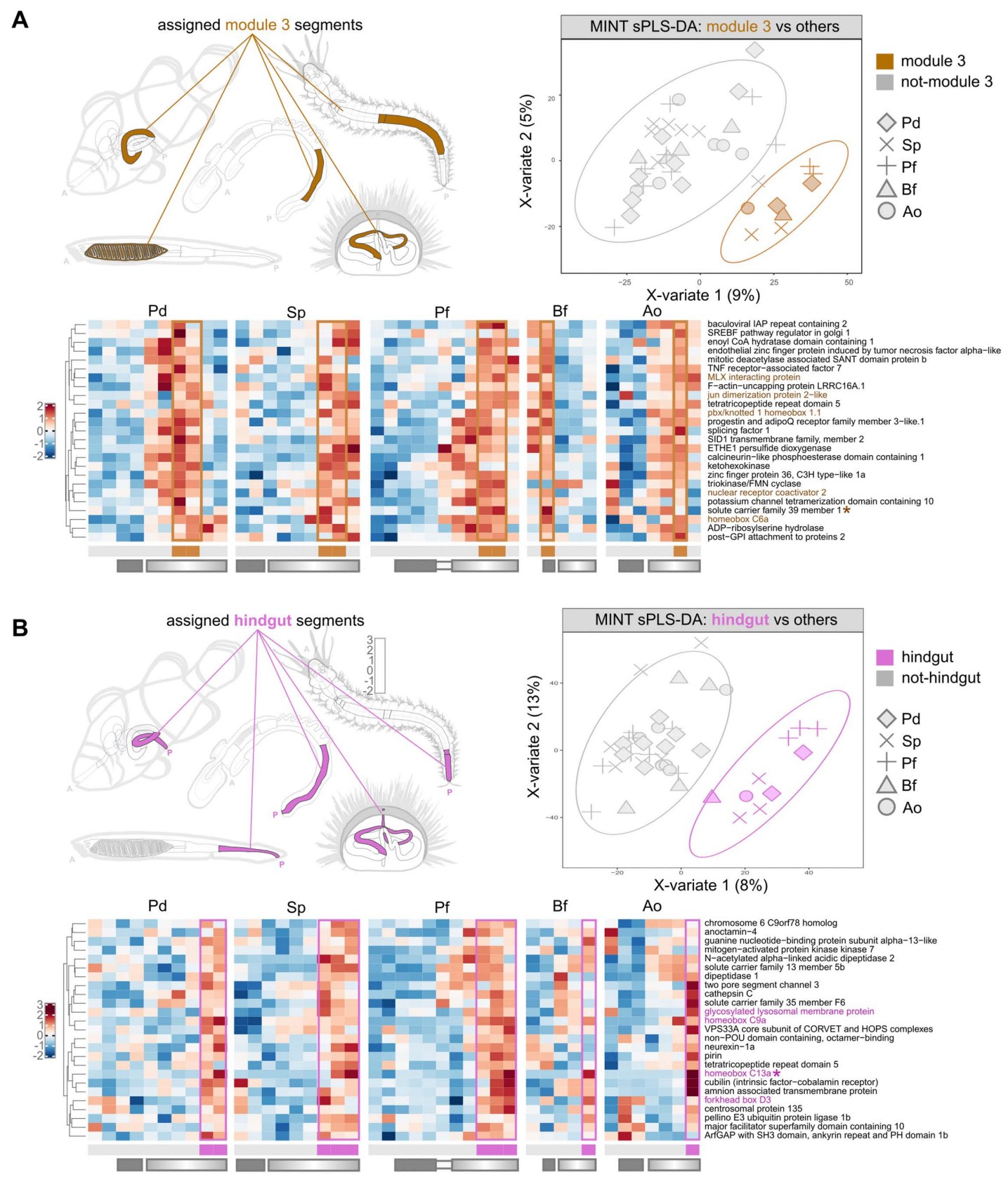

**Fig 7. Module 3 and Module 4 sPLS-DA. A)** sPLS-DA of bilaterian gut segments marked by Module 3 TFs. **Left:** illustration of the segments to be discriminated (Module 3 segments). **Right:** separation of segments according to sPLS-DA discriminant genes. **Bottom**: For each species, heatmap showing the expression pattern (z-scores) of the top discriminant genes identified through sPLS-DA. Genes (rows) are clustered according to hierarchical clustering. Segments (columns) are ordered according to AP position. Solid and graded rectangles under the heatmaps indicate segments previously assigned to block and gradient gut compartments, as defined in previous sections. Gene names followed by an asterisk constitute the optimal sufficient discriminant set. Gene names in color are TFs. **B)** sPLS-DA of the most posterior bilaterian gut segments (also marked by Module 4 TFs). **Left:** illustration of the segments to be discriminated (Module 4 segments). **Right:** separation of segments according to sPLS-DA discriminant genes. **Bottom**: for each species, heatmap showing the expression pattern (z-scores) of the top discriminant genes identified through sPLS-DA. Genes (rows) are clustered according to hierarchical clustering. Segments (columns) are ordered according to AP position. Solid and graded rectangles under the heatmaps indicate segments previously assigned to block and gradient gut compartments, as defined in previous sections. Gene names followed by an asterisk constitute the optimal sufficient discriminant set. Gene names in color are TFs. The data underlying this Figure can be found in https://doi.org/10.5281/zenodo.17746910.

Overall, our results strongly support the deep conservation of this gut function, not only as one of the main evolutionary-conserved pan-bilaterian gut modules, but also as one with a generally conserved relative anteroposterior position. It puts forward a set of markers that may in fact be ancestral components of the bilaterian gastrointestinal endocrine system, and hint at the type of nutritive stimuli it may have been responsive to. We suspect that loss in Bf, and gill segment expression, may be explained by the amphioxus-specific Hatschek's pit, involved in endocrine regulation [97].

**Module 4/hindgut.** One of the most striking observations about the modules we identify is the extreme scarcity of shared markers of hindgut segments, even as a result of omics level investigations. Not only do adult bilaterian hindguts only share the expression of 3 TF orthogroups (when including *cdx*), these are the three extremely well-known markers of the embryonic hindgut [20,36,40,52,98–101]. We wondered whether a *supervised* dimensionality reduction analysis could possibly expand such a signature. Accordingly, we performed sPLS-DA coercing all posteriormost segments to belong to shared identity, here based on their shared posteriormost anatomical position (Fig 7). Strikingly, sPLS-DA identifies the sole Hox PG011-15 as the optimal gene set sufficient to discriminate the hindgut from other segments. Within the expanded sPLS-DA signature (of genes that mark the hindgut but not necessarily exclusively, Fig 7), we only recover one other classic TF, *foxD*. *FoxD* is indeed another extremely well characterized regulator of hindgut function in vertebrates [102], a key component of the larval hindgut gene regulatory network of sea urchin [48], is known to be expressed in the hindgut endoderm in the hemichordate *Saccoglossus kowalevskii* [103] and in amphioxus [104,105], and is generally associated with the early development of deuterostome hindguts (see [20], and references within). We here also recover *foxD* in the posterior segments of a protostome species (as in the larval stages of others, see, e.g., [106]). Yet, based on both the expression patterns we recover here (e.g., Pd, Sp), and the ontogenetic shift described for Sp in Fig 3A, we would classify this TF as a "terminal" marker rather than a specific posterior marker at the pan-bilaterian level. We therefore confirm that posterior Hox and posterior ParaHox genes may be the only pan-bilaterian markers of the through-gut posterior.

An extended set of bilaterian hindgut markers may have to be sought in the conservation (or convergence) of non-TFs expression patterns. Within the expanded sPLS-DA hindgut signature, we notably recover *amnionless*, *cubilin*, *cathepsin C*, *glmp/Ncu-g1*, *slc35f6*, *vps33a*, many of which may serve as broad markers of the bilaterian hindgut. We note that many of these components belong to the cubam/endocytosis system. In teleost, posterior expression of *amnionless/cubilin* has been shown to correspond for the most part to a specialized class of enterocytes responsible of direct protein uptake by endocytosis ("lysosome rich enterocytes," [107–109]). This same function/cell type is also believed to be also present in invertebrate chordates [24,27] and *cubilin/amnionless* has also been recently characterized as localized to the hindgut of *Ciona intestinalis* [110]. Our results suggest that this same cell type, or at least the posterior presence of an endocytic intestinal function, is a deeply conserved through-gut feature, and in fact may be the only conserved function of the terminal section of bilaterian guts. The conservation of hindgut regionalization of such components across body plans that had their last common ancestor hundreds of millions of years ago suggests that the (posterior) regionalization of endocytic

activity may have been a key feature of through-gut function in the bilaterian ancestor. Still, we note that these functions are not exclusive to, nor necessarily peaking in, the posteriormost segment. We consistently fail to detect conserved bilaterian features exclusive to bilaterian posteriormost segments, beyond anatomical position and the expression of posterior Hox and ParaHox genes.

## Discussion

To better understand how to interpret bilaterian through-guts, we collected anteroposteriorly-ordered transcriptome data of the adult gut tube of five phylogenetically informative species: an annelid, a sea urchin, a hemichordate, a cephalochordate, and a vertebrate. Overall, our cross-species comparison highlights that the highly divergent and heterogeneous gene expression patterns of adult guts across species appear to develop within a conserved, bipartite global gene expression structure, and still retain a number of modules with overall conserved relative AP position (summarized in Fig 8A and detailed in S8 Fig). Crucially, such a bipartite global organization matches a conserved, bipartite, adult expression of Hox genes.

### Adult through-guts display "larval" AP-signatures

Intriguingly, the modules found to conserve relative AP information across species are largely represented by well-studied embryonic/larval markers. Hand-in-hand with this observation is our finding that the AP order of expression of classic, bilaterian-conserved, AP embryonic markers is also a feature of adult guts, to the point that such TFs may well be considered adult AP markers too. Far from the phylotypic stage, adult bilaterian through-guts thus appear to maintain an open window to early embryonic/larval patterning systems (Fig 8B), even in species whose metamorphosis drastically restructures the gut tube itself [79]. These markers must therefore be maintained across ontogeny or re-deployed in the adult gut. To this point, studies that have most recently directly addressed the bridge between gene expression patterns before and after metamorphosis, are indeed cementing the re-deployment of embryonic and larval molecular signatures into juvenile life as a feature of most cell types, including those of the gut ([78], *Paracentrotus lividus*). We show here that not only are these genes expressed across the metamorphic gap, but they are so in analogous AP order that allows direct correspondences between the domains marked at each life stage. Furthermore, we show that this order is evolutionarily conserved, allowing to further draw equivalent correspondences across adult guts of different species. Our results are particularly interesting in the context of the ancient discussion on the origin of larvae and the evolution of metamorphosis in bilaterians, a debate that has animated the literature since more than a century [reviewed in 111]. Our findings that embryonic patterning genes are redeployed in adult guts and maintain similar anteroposterior positioning highlight an unappreciated continuity across metamorphosis, even in cases characterized by dramatic morphological transformations, and bring forward a striking equivalence between larval and adult gut patterning systems. From a "larva-first" point of view, our findings would suggest that metamorphosis does not erase the early program but rather reuses it within a new context. In that sense, metamorphosis could be seen not as a rupture, but as a reconfiguration of a conserved regulatory logic, inherited from the common bilaterian ancestor [112]. At the same time, our identification of adult bilaterian guts as AP-patterned systems (i.e., structures patterned by the same systems that are seen in larval guts) brings forward the alternative possibility: that what is deployed in larvae is an ancestral adult programme. Though our data cannot formally exclude either theory, our findings bring new elements to fuel this long-standing debate, specifically in highlighting the fact that bilaterian adult through-guts display an AP-patterning system mostly considered to be a larval exclusive.

### Adult functions of AP-patterning genes

Though the adult display of such expression patterns could intuitively be explained as a mere leftover of a deeply constrained, conserved, and ancestral embryonic gut patterning system, the conservation of their adult expression in all species considered, across hundreds of millions of years of evolution, additionally points towards a key role played by

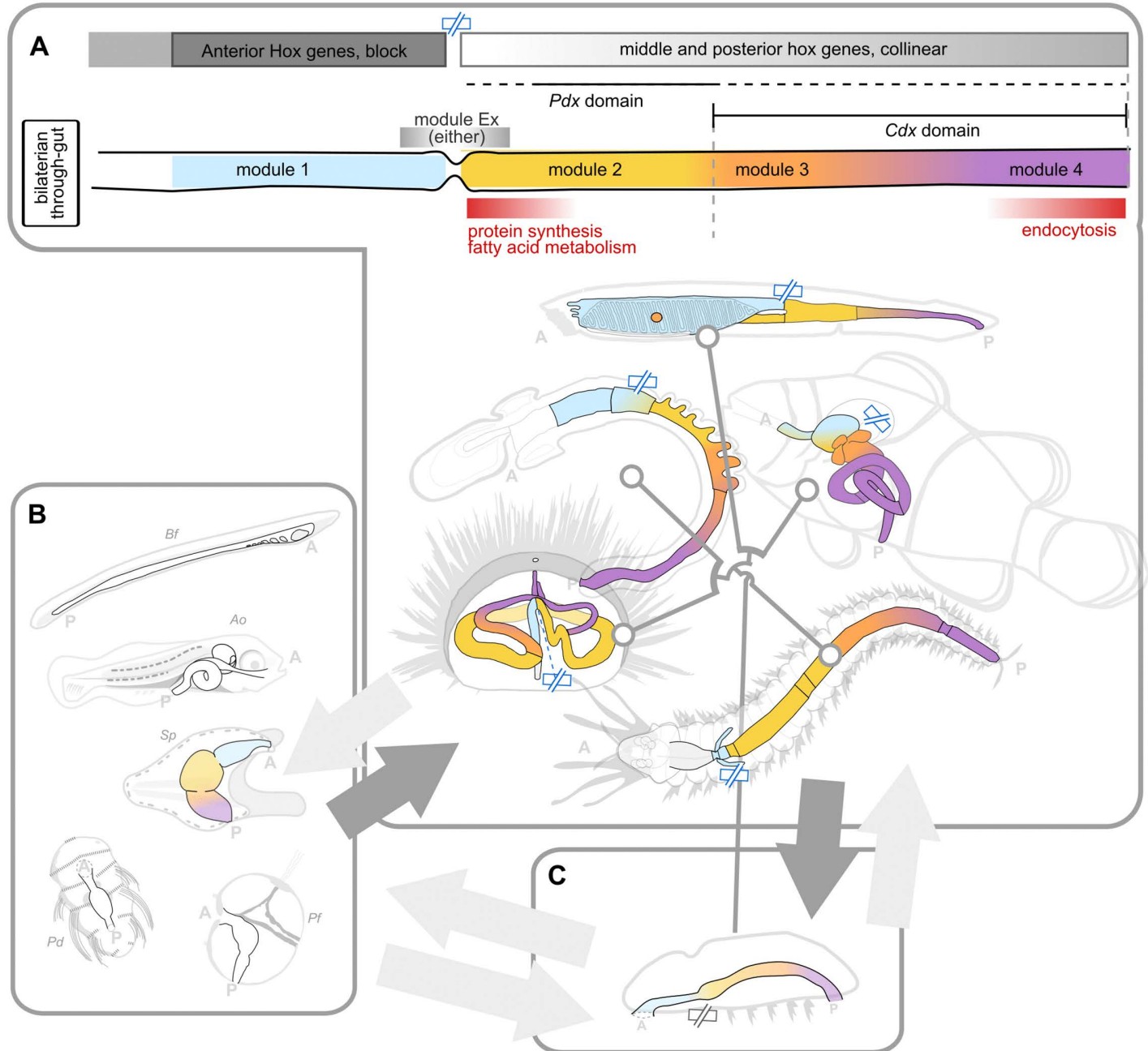

**Fig 8. Summary and conclusion figure. A)** Summary model of the conserved elements of the adult bilaterian through-gut, and their AP distribution, as identified in this work. Modules are color-coded. Slashed rectangle pictogram (blue, topmost): boundary between "block" and "gradient" compartments (i.e., "transition sphincter") as defined in the main body of text. **B)** Larval bilaterian through-guts. A correspondence between larval and adult AP gut markers (dark arrow) was identified for sea urchin (Sp). The AP patterning of adult bilaterian guts may in turn be a conserved feature of larval through-guts (light arrow). **C)** Reconstruction (dark arrow) of the possible configuration of the ancestral through-gut configuration, which gave rise to the diversi-fication of adult guts seen in extant species (light arrow to panel A). Our interpretative lens would further provide an investigative angle to approach the relationship between the ancestral through-gut configuration and that of its larva and the larvae of extant species (light arrows to and from panel B). A: anterior, P: posterior.

these genes in the adult. Could embryonic gut patterning genes be genes with key adult functions as regulators of adult endodermal functions, gut "terminal selectors" (using here "terminal" in its meaning of "terminal differentiation")? Certainly, these genes are expressed in adult guts spanning a variety of architectures and dietary specializations, and they are clearly redeployed even in cases of ontogenetic substrate discontinuity with their patterned embryonic correspondent. Much is speculative about the configuration of the gut of the bilaterian ancestor, and even more so about its relationship with that of its embryonic form. We here add an additional layer to the question by demonstrating that the gene expression patterns of adult and embryonic/larval guts have a non-trivial continuity, implicating the adult digestive physiology and ecology of bilaterian ancestors among the elements informing the AP patterning mechanisms of the bilaterian gut. To what extent does the permanence and function of these genes in adult bilaterian guts affect the selective constraints on the patterning processes of their embryonic origin? Incidentally, this continuity between adult and embryonic gene expression patterns suggests that the strong degree of conservation of this AP system across adult bilaterian through-guts may in turn mirror a conservation of the pattern at their embryonic stages. We can only show such an actual correspondence based on published sea urchin larval data [37]: dedicated characterizations of the AP pattern of expression in the larval or embryonic forms of the other species will reveal whether such likely conservation [38,77] holds true.

## From a conserved Hox scaffold to highly divergent transcriptional profiles

Though we focused on establishing a comparative basis among bilaterian guts, our cross-species comparison in fact highlights, above all, the extreme heterogeneity of gene expression in adult guts of different species, of course reflecting the remarkable variety of gut forms and functions across phyla, species, and diets. At the pan-bilaterian level, gut segments at equivalent AP positions generally cluster poorly. Regardless, such a remarkably flexible deployment of genes and functions invariably appears to be organized across a two-compartment scaffold that is comparable across species. That is, we see a cross-species conservation of global AP spatial expression *structure*, rather than relative AP *localization* of most of the individual genes themselves. In the case of adult guts, this bipartite global organization is underlaid by a bipartite deployment of Hox genes, a deployment pattern observed in the embryos of model vertebrate species [74–76] but which we here extend to the adult, and generalize across bilaterians. Such a bipartite deployment of Hox genes would therefore also be the most likely condition of the adult gut of the bilaterian ancestor.

## Modules and modularity of the bilaterian through-gut

Among the overall heterogeneity of individual gene expression patterns, we were able to extract conserved sets of TFs that do nonetheless appear to conserve relative AP expression across species, and which we initially group into five modules. Our analysis suggests that if cross-species correspondences are to be found at a more granular level than that offered by "block" and "gradient" compartments, these five modules could be useful anchoring points and cross-species landmarks. The identification of conserved AP patterns of expression across extremely divergent guts and across hundreds of millions of years of evolution further identifies such modules as key feature of the bilaterian guts and suggests that the relative AP sequence recovered in the three non-chordate invertebrate species may have been the most likely configuration of the ancestor through-gut (Fig 8C). We highlight, for example, the Module 3 as a putative "sensory" neuro/endocrine function possibly relating to bilaterian pancreatic functions, and the overall paucity of markers (Module 4) able to support the hindgut at the pan-bilaterian level. Our analysis identifies each of these modules, and their markers, as prime leads and focal points for future investigations, and identifies the sphincter marking the boundary between the two Hox compartments as a key anchoring point for cross-species comparisons, and for gut Eco-Evo-Devo in general. Notably, our investigation of expression patterns of species at key phylogenetic positions allows the preliminary reconstruction of evolutionary trends that specifically highlight the vertebrate gut as a highly divergent gut architecture. Indeed, we consistently note a strong conservation of AP patterns of expression between the adult gut of *Platynereis* and of the two ambulacrarian species and increasing degrees of reorganization in the chordate branch. By identifying the Hox

compartment boundary as a key transitional point of bilaterian gut organization, our interpretative framework allows us for the first time to put forward evolutionary hypotheses formulated in terms of shifts and rearrangements with respect to these invariant features. That is, in terms of AP shifts of modules with respect to a fixed Hox compartmentalization or, vice versa, in terms of shift of Hox compartments and their boundary with respect to the fixed AP measure of the adult gut.

We put forward this work and data not only because it provides a still rare resource of AP-resolved whole-transcriptome data of adult guts across organisms at key phylogenetic positions (joining most recent precious efforts, [52,64–66,113–124]), but also because discriminating between derived and conserved features of such an important and heterogeneous organ as the through-gut remains central to almost all Eco, Evo, and/or Devo investigations that touch on this organ system. We here provide a hypothesis for the AP patterning system of the through-gutted ancestor and its possible functions and also propose an interpretative lens we have found to best reflect the organization of extant bilaterian guts. We believe that the application of such an interpretative lens, and the testing of this model against open questions of bilaterian and non-bilaterian through-gut biology will provide rich insight across all three fields of Eco-Evo-Devo.

## Materials and methods

### Animal rearing licences/ethics approval

Clownfish were reared under licence number 21-04-1658 issued from Academia Sinica's Institutional Animal Care and Use Committee (IACUC) to the Marine Eco-Evo-Devo Unit at the Linhai Marine Research Station. Experiments (dissections and sample collection) were performed within the provisions of the same licence. Rearing and sample collection of invertebrate species (*Branchiostoma floridae*, *Strongylocentrotus purpuratus*, *Ptychodera flava*, and *Platynereis dumerilii*) is exempt from IACUC oversight, as per institutional policy.

### Animal rearing methods

**Amphiprion ocellaris.** Young adult *A. ocellaris* anemonefish were bought from a commercial anemonefish farm (S.T Biotechnology , Changhua County, Taiwan) and reared under standard conditions analogous to what described in Roux and colleagues [125]. Specifically, young adults were housed in a 70L tank in recirculating natural seawater under a 15 h:9 h day–night photoperiod, at 28 °C. Fish were fed thrice daily either with dry pellets (Hikari Premium Megabite; Kyorin Food Industries) or with fresh seafood. Fish were not fed on the day of collection to minimize gut contents and possible RNA sample contamination.

**Branchiostoma floridae.** Amphioxus (*Branchiostoma floridae*) adults were originally collected from Old Tampa Bay (Florida, USA) and subsequently transported to the Institute of Cellular and Organismic Biology (ICOB), Academia Sinica, Taiwan, for laboratory rearing. Animals were maintained in plastic culture boxes supplied with natural habitat sand and aerated with air pumps. They were fed a mixed algal diet consisting of *Isochrysis galbana* (Pingtung, Taiwan), *Tetraselmis chuii* (Pingtung, Taiwan), *Rhodomonas lens* (USA), and *Rhodomonas* sp. (Japan). The rearing protocols were adapted from Yu and Holland [126] and Yong and colleagues [127].

**Ptychodera flava.** Adult *Ptychodera flava* acorn worms were collected from intertidal ridge of Chito, Penghu, Taiwan, and kept in aquaria as previously described [128].

**Strongylocentrotus purpuratus.** Adult *Strongylocentrotus purpuratus* were provided by Pete Halmay and Amro Hamdoun (Scripps Institution of Oceanography, University of California, San Diego). Animals were maintained at 15 °C and fed a diet of kelp (kombu).

**Platynereis dumerilii.** *Platynereis dumerilii* belonged to the "Heidelberg" strain originating from the Mediterranean Sea, and were cultured at the ICOB, Academia Sinica, as described in Kuehn and colleagues [129]. The animals were maintained under a light/dark photoperiod of 16:8 hours with 8 days of simulated moonlight every 28 days and were checked for death and premature maturation three times per week. Premature worms were separated into male or female maturation bowls. The worms were fed three times a week with a varied diet consisting of spinach (spinach florets; Sin

Mei company, Changhua County), algae (*Tetraselmis chui*; Algae-Supply company, Taitung), spirulina (Omega; Golden Prawn Enterprise Co., Kaohsiung), and fish food (Tetra Bits Complete; Tetra GmbH, Germany). Twenty mL of food were given to each box each day. Natural seawater used for culturing was filtered through 1, 5, and 0.22 μm in the lab.

## Dissections and sample collection

A summary illustration of the dissection process for each organism is provided in S1 Fig.

**Amphiprion ocellaris.** Three *A. ocellaris* young adults at best health were each collected from the rearing tank and immediately euthanised by transfer to a cold bath of overdosed tricaine methanesulfonate (MS-222, ethyl 3-aminobenzoate methanesulfonate salt, tricaine mesylate; [Merck/Supelco CAT#A5040, 200 mg/L] dissolved in ice-cold seawater). To guarantee tissue freshness and to try to prevent the establishment of artefactual transcriptional responses, each new fish was collected only after full dissection of the previous one, and tissues were kept covered by ice-cold seawater throughout the dissection. For dissection, each fish was pinned left-side down to an ice-cold dissection mat, through the eye and through the vertebrae posterior to the trunk bar. After removing the operculum, a lateral window was cut-out to expose the visceral cavity. The cavity was rinsed and filled with ice-cold seawater, and the liver was collected by severing the duct connecting it to the anterior intestine (usually best accessible from the left side). The entire digestive tract was then excised by cutting around the anal opening on one end, and pulling on the gill arches at the other. The digestive tube was then unfolded by carefully tearing apart the tissue between intestinal folds and around the anterior intestine. After trimming away the gill arches (anterior), any carry-over anal dermis (posterior), as well as all contaminating tissues, the digestive tract was further subdivided into 6 segments defined as follows (anterior to posterior): esophagus (up to stomach sphincter); stomach (anatomically distinct); pyloric caeca (anatomically distinct); anterior intestine (up to first noticeable constriction and change in opaqueness; the middle intestine was usually transparent); middle intestine (up to next noticeable constriction); posterior intestine. Note that the gallbladder, attaching to the anterior intestine segment, was preemptively punctured to empty it of its contents. Also note that the pancreas could not be distinguished/dissected as a separate tissue and remained likely associated to the anterior and mid intestinal segments.

Each newly-obtained segment was transferred to a dedicated sterile 2 mL microcentrifuge tube filled with 750 mL ice-cold Trizol (TRI Reagent; Merck/Sigma-Aldrich CAT#T9424) containing three autoclaved stainless-steel beads (EBL Bio-technology CAT#SB2006), and kept on ice until the end of the dissection (less than 15 min total for each set). After each set of dissections, samples were homogenized by mechanical agitation using a vibrating-bead mill (TissueLyser II, Qiagen CAT#85300, RRID:SCR_018623; 3 min, 30 Hz, room temperature). Homogenized sample lysates were stored at −80 °C overnight until RNA extraction the following day.

**Branchiostoma floridae.** Adult amphioxus *B. floridae*, each of 2.5–4 cm in length, were anaesthetized using MS-222 (Merck/Supelco CAT#A5040; 160 ppm in filtered seawater, adjusted to pH 7.5–8.0) for at least 10 min before dissection. One animal was dissected at a time, placed on a glass petri dish, with a few milliliter of filtered seawater added to prevent organs from drying out. The atrium and ventral-lateral portion of metapleural folds were cut open to expose the digestive tract. Based on prominent anatomical features, we designated the digestive tract into five "organs": the gill, endostyle, midgut, liver (hepatic cecum), and hindgut. These names are used for practical purpose and do not imply their evolutionary homology with organs of other animals.

**Ptychodera flava.** Acorn worms were anesthetised in 0.08% of MS-222 in filtered seawater for 20 min [130]. Due to strong adhesion of the acorn worm's digestive tract to the body wall, complete isolation of the gut was not possible to the best of our efforts. Accordingly, the whole cross-section of the worm (including enveloping body wall tissues) was amputated according to the internal endodermal features and including the proboscis region with stomochord, the collar region with the mouth, and the trunk region with pharynx, esophagus, hepatic region, and intestine.

**Strongylocentrotus purpuratus.** Adult sea urchins were anesthetized in 0.8 g/L MS-222 for 20 min, then dissected along the ambulacral axis from the mouth, resulting in two separate portions of the body. The smaller portion contained

the esophagus and anterior stomach (2e–3s–4s) as a continuous tissue, and a separate posterior intestine fragment (9i). The larger portion included the remaining digestive tract (5s–6s–7i–8i). Continuous tissue regions (3s–4s, 5s–6s, 7i–8i) were divided equally for downstream analysis. The pharynx (1p) was also isolated from the teeth. All dissections were carried out on filtered seawater maintained on ice to preserve tissue integrity and the isolated tissues were immediately snap-frozen in liquid nitrogen. Particular attention was given to the anterior stomach (3s), which is rich in digestive enzymes and thus particularly susceptible to RNA degradation.

**Platynereis dumerilii.** Adult *Platynereis dumerilii* specimens, reaching 60–70 segments and have not yet started the pre-mature stage, were dissected to isolate the digestive tract. The worms were anesthetised in plastic plates filled with 7.5% $MgCl_2$, as per [129]. Dissection was performed on the ventral side from anterior to posterior. Morphological landmarks were used to define 10 segments along the AP axis. These 10 segments correspond to the mouth, stomodeum, transition between stomodeum and esophagus, transition from esophagus to stomach, stomach, transition from stomach to midgut, midgut, transition from midgut to hindgut, and hindgut. We endeavored to isolate specific tissues with high precision; however, the mouth and hindgut regions are consistently affected by an inseparable layer of skin epidermis.

### RNA extraction

**Amphiprion ocellaris.** On the day of extraction, sample lysates (in Trizol) were thawed on ice and centrifuged 10 min in a tabletop microcentrifuge at 11,000 g, 4 °C to collect any leftover tissue debris to the bottom. 750 µL of each supernatant was loaded into a dedicated RNA extraction column (NucleoSpin RNA Mini kit; Macherey-Nagel CAT#740955.50) and processed according to manufacturer recommendations. Specifically, 350uL of 70% Ethanol (Honeywell/Riedel-deHaen CAT#32221, in $ddH_2O$) were added to the filtered flowthrough, the solution was then vortexed, and loaded onto a RNA binding column. After desalting (kit-supplied Membrane Desalting Buffer), contaminating DNA was digested by a 15 min incubation in recombinant DNAse (in Reaction Buffer, room temperature). Guanidine hydrochloride/ethanol wash buffers (RAW2 and then, twice, RA3) were then sequentially used to denature proteins, discarding the flowthrough from centrifugation after each step. After a further centrifugation to remove all possible leftover buffer/ethanol (2 min, 11,000*g*, 4 °C), membrane-bound RNA was eluted in 30 µL of RNAse-free water (UltraPure DNase/RNase-Free Distilled Water; Invitrogen/Thermo Fisher Scientific CAT#10977015), in a 1.5 mL RNAse-free tube (kit-supplied). RNA amounts and quality were initially assessed with a NanoDrop spectrophotometer (NanoDrop Lite; Thermo Scientific CAT#840281500) and by agarose gel electrophoresis (2:1 intensity ratio of 28S:18S bands; 800 ng of sample per lane, 0.6% Agarose in TAE-buffer, 1 h, 50 V). Extracted RNA was stored at −80 °C until sequencing.

For samples that showed low purity (Liver03, Pyl01, Ain01; A260/A280 ratio < 1.8) an additional purification step was performed. Accordingly, eluted RNA was diluted to 100 µL total volume with DEPC water, and mixed with an equal volume of phenol:chloroform (from GeneRacer Kit, Invitrogen/Thermo Fisher Scientific CAT#L150201) by vigorous vortexing for 30s. Phases were separated by centrifugation at maximum speed on a tabletop minicentrifuge (21*g*, 5 min, RT), and the aqueous phase was collected into a new tube. To re-precipitate the RNA, 12 µL of a 10 mg/mL mussel glycogen, 3M sodium acetate pH5.2 solution were added (all from GeneRacer Kit, Invitrogen/Thermo Fisher Scientific CAT#L150201). After further addition of 220uL of 95% ethanol, the solution was vortexed briefly. Samples were stored at −80 °C overnight to allow flocculation. The following day, samples were centrifuged (21*g*, 20 min, 4 °C) and all supernatant was removed paying attention to preserve the RNA pellet. The pellet was then washed by adding 500 µL of 70% ethanol, followed by manual inversion of the tube and brief vortexing. The sample was then centrifuged 2 min, 21*g*, 4 °C, the ethanol supernatant was removed, and the sample was centrifuged again under the same settings. Again, any remaining ethanol was removed. Pellets were air-dried for 15–30 min, or until becoming vitreous/transparent, and then resuspended in 30 µL DEPC water. RNA amounts and quality were re-assessed with a NanoDrop spectrophotometer, usually showing minimal RNA loss compared to pre-cleanup amounts, and an increase of around +0.5 points in their A260/A280 ratio. Cleaned-up RNA was stored with the rest of the samples at −80 °C until sequencing.

**Branchiostoma floridae.** Immediately after isolation, each organ was immersed in 350 µL of ice-cold "RLT" buffer (RNeasy Mini kit, Qiagen) containing β-mercaptoethanol (RLT/bME, 100:1 v/v). The sample was vortexed 15 s and placed on ice. Once all target organs were collected and treated in RLT/bME, the samples were homogenized using a BioMasher II disposable homogenizer and a battery-powered grinder PowerMasher (Nippi.; 10–30 s, or until no visible tissue clumps remained). The homogenates were kept in the grinding tube and stored in −80 °C for a few days before RNA extraction.

RNA was extracted using the RNeasy Mini kit (Qiagen CAT#74104). Frozen samples in RLT/bME were thawed on ice for 5–10 min. We followed the manufacturer's protocol for RNA extraction, with the exception of extending the DNase treatment to 30 min at room temperature. Finally, RNA was eluted in 16 µL of nuclease-free water to obtain a concentrated RNA solution.

**Ptychodera flava.** To optimize RNA quality, dissected samples were immediately rinsed in cold RNAlater Stabilization Solution (Invitrogen/Thermo Fisher Scientific CAT#AM7020) to clear any remaining digestive enzyme in the gut [131]. These stabilized samples were then homogenized in TRIzol with BioMasher II disposable homogenizer and a PowerMasher (Nippi.) on ice. The RNA extraction of tissue lysate was performed with Direct-zol RNA Miniprep Kit (Zymo Research CAT#R2050) after a pre-filtering step with QIAshredder biopolymer-shredding system (Qiagen CAT#79656).

**Strongylocentrotus purpuratus.** Snap-frozen tissue samples were ground in liquid nitrogen to a fine powder. The powdered samples were weighed and lysed using TRIzol Reagent (Invitrogen/Thermo Fisher Scientific CAT#15596026). RNA was then extracted using the Direct-zol RNA Miniprep Kit (Zymo Research CAT#R2050), according to the manufacturer's instructions.

**Platynereis dumerilii.** Immediately after collection in a 1.5 mL Eppendorf tube, gut segments were placed on ice to preserve RNA integrity, with 100 µL TRIzol reagent (Invitrogen/Thermo Fisher Scientific, CAT#12183555). Gut tissue was then homogenized using a disposable pestle and a battery-powered homogenizer. Up to 1 mL of TRIzol reagent was then mixed thoroughly into each sample. After a 5 min incubation at room temperature, Chloroform was added at a fifth of a volume, samples were shaken vigorously for 15 s, and incubated at room temperature for 2–3 min. After centrifugation (12,000$g$, 15 min, 4 °C) the upper 2/3 aqueous phase was carefully removed. RNA was precipitated by addition of half a volume of 100% isopropanol and gentle mixing by tube inversion. Samples were incubated 10 min at room temperature, centrifuged (12,000$g$, 10 min, 4 °C), and the supernatant was removed. RNA pellets were washed with 75% ethanol-ddH$_2$O and vortexed briefly. After a further repelleting of the RNA (7,500$g$, 5 min, 4 °C) and removal of all supernatant ethanol, pellets were air-dried for 10–15 min. As a final step, pellets were resuspended in 30 µL DEPC-treated water. RNA concentration and integrity were measured using a NanoDrop spectrophotometer.

### Sequencing

**Amphiprion ocellaris.** Quality-control, library preparation and sequencing of extracted RNA were performed by the High Throughput Sequencing Core hosted in the Biodiversity Research Center at Academia Sinica, Taipei. RNA-Seq libraries were generated from total RNA using the Illumina Stranded mRNA Prep mRNA Sample Preparation Kit with UDI indices (Illumina, USA) according to manufacturer's instructions. Surplus PCR primers were removed using AMPure XP (Beckman Coulter Life Sciences, USA). Final cDNA libraries were checked for quality and quantified using Qubit (ThermoFisher Scientific, USA) and Fragment Analyzer for size profiling (Agilent, USA), and concentration normalized using KAPA Library Quantification Kit for Illumina Platforms (Roche, USA). Sequencing was performed on an Illumina NextSeq2000 for paired-end 150 base format. Libraries were loaded in a P2 flow cell.

**Branchiostoma floridae.** Extracted RNA was initially quantified with a NanoDrop spectrophotometer followed by quantification with a Qubit Fluorometer. RNA integrity (RQN) was assessed using the BioAnalyzer (Agilent). When five RNA sub-samples from different organs of the same animal each contained at least 200 ng of RNA and displayed high quality (RQN 8–10), these samples were selected for library preparation using the TruSeq stranded RNA-polyA method.

The multiplexed cDNA libraries were then sequenced using a Rapid Run flow cell on a HiSeq 2500 sequencer (Illumina) with paired-end 151 bp reads.

**Ptychodera flava.** Total RNA as input for Illumina Stranded mRNA Prep mRNA Sample Preparation Kit with UDI indices (Illumina, USA) according to manufacturer's instructions. Sequencing was performed on an Illumina NextSeq2000 for paired-end 150 base format. Libraries were loaded in the P3 flow cell. The fastQ files were generated and demultiplexed using bcl2fastq v2.20 pipeline.

**Strongylocentrotus purpuratus.** RNA-seq experiments were conducted using four biological replicates. Total RNA quality and concentration were assessed using the Fragment Analyzer and Qubit Fluorometer. Bulk RNA sequencing was performed on the Illumina NextSeq2000 platform, generating 150 bp paired-end reads with high sequencing depth to ensure transcriptome coverage and quantification accuracy.

**Platynereis dumerilii.** The quality of the extracted RNA is assessed using Agilent Fragment Analyzer 5200 to check the integrity and Qubit 4 Fluorometer to check the concentration. The experiments are done in three biological replicates. Bulk RNA-seq are performed using Illumina Hi-seq 2500 with 125 bp paired-end reads deep sequencing which were performed by the NGS High Throughput Genomics Core in Academia Sinica.

## Clownfish Hox complement reconstruction

The reconstruction of *Amphiprion ocellaris* Hox genes, their numbers and their identity, was based on the gene models available from both the NCBI reference annotation (ASM2253959v1) and the Ensembl annotation at the time (AmpOce1.0, now matched to ASM2253959v1), taking also into account the relative position of the genes within each cluster. Furthermore, only genes for which the corresponding protein was predicted to contain a homeobox domain were maintained. The resulting list of assigned identities was then cross-compared with the expected percomorph configuration based on Hox cluster evolutionary patterns in teleosts [132], and was indeed found to be most similar to the configuration in the closest reference species *Oryzias latipes* [133] and *Oreochromis niloticus* [134]. We conservatively define a final set of 47 Hox genes for *Amphiprion ocellaris*.

## Orthology assignment

A complete list of the protein IDs of all annotated proteins from each species were extracted from the corresponding Gene Transfer Format (.gtf) files downloaded from NCBI, referring to the following published annotations: Ao: GCF_022539595.1; Bf: GCF_000003815.2; Pf: GCF_041260155.1; Sp: GCF_000002235.5. Custom files, based on GCA_026936325.1, were used for Pd. The protein ID corresponding to the longest transcript isoform of each (protein-coding) gene was selected as representative entry (deduplication). These IDs were then used to subset the full proteome of each species, as downloaded from NCBI, using the "filterbyname" shell script of BBTools v39.01 (RRID:SCR_016968; Bushnell B., http://sourceforge.net/projects/bbmap/). The resulting deduplicated proteomes were used for orthogroup identification using the standard Orthofinder2 pipeline [86] with default setting. Anticipating the potential needs of indexing the generated ortholog table to known vertebrate homologs, the deduplicated proteomes from human (*Homo sapiens*, Hs), mouse (*Mus musculus*, Mm), and zebrafish (*Danio rerio*, Dr) were also included in the orthology reconstruction (Hs: GCF_000001405.40; Mm: GCF_000001635.27; Dr: GCF_000002035.6).

## Hox orthogroup reconstruction

We note that Orthofinder fails to reconstruct coherent orthogroups for the Hox paralogues across bilaterians due to the complex evolutionary history of this gene family, often marked by independent duplication of common ancestors across lineages [135,136]. In our analysis, this resulted for example in the fragmentation of all Hox genes into 21 different orthogroups, with paralogues at equivalent Hox positions split across different orthogroups. In the perspective of our comparative analysis, we are interested in comparing the expression pattern of Hox genes at equivalent positions

rather than necessarily preserving true phylogenetic relationships. Accordingly, all 21 orthogroups containing Hox genes were discarded and replaced by a manually defined final set of 8 "Hox orthogroups" ("OG000PG01," "OG000PG02," "OG000PG0304," "OG000PG05," "OG000PG0607," "OG000PG08," "OG000PG0910," "OG000PG1115") grouping Hox genes at equivalent cluster positions across species (e.g., "OG000PG01" grouping all Hox genes at position 01 of each Hox cluster). In cases where a given cluster member was absent from at least one species, which would prevent the use of all other species' matches in comparative analysis, a merged "Hox orthogroup" was created by combining paralogues from neighboring positions (e.g., "OG000PG0304" grouping all Hox genes at positions 03 and 04 of each Hox cluster, due to the absence of Hox4 in Sp). Note that for the special case of Sp Hox genes, which break genomic collinearity [70], the three 5′ Hox genes were associated to the first three Hox orthogroups, therefore prioritizing phylogenetic correspondence. We observe that their expression within the gut is indeed anterior.

## Transcription factor identification

To identify TFs, we followed AnimalTFDB protocols and performed biological sequence analysis on each species' dedu-plicated proteome using profile hidden Markov Models (HMMER) with InterPro's Pfam-A models and AnimalTFDB v4.0 self build HMM files [137–139]. With these methods, 1778, 1933, 1339, 992, and 1425 TFs were identified in Ao, Bf, Pf, Sp and Pd, respectively. When each species' TFs were summarized at the Orthogroup level, we noticed that members of a same orthogroup would sometimes be classified as TFs in one or more species, but not in other ones. To resolve this mismatch, we decided to apply an inclusive approach to TF definition by further including as TFs all "non-TFs" falling within the same orthogroup of another species' TF (i.e., we expanded the TF list based on domain-search alone). After adjustment for Hox gene orthogroups (see section "Hox orthogroup reconstruction"), we thus obtain a total of 1426 TF orthogroups as a basis for comparative analysis. Due to the constraints of cross-species comparisons, and the use of Orthogroups, we stress that our final list of TF orthogroups does not include species-specific TFs (i.e. TFs with no ortho-logues in any other species), regardless of any fundamental role they may well play within the adult gut of their particular species.

## Analysis of bulkRNAseq data

**Raw data preprocessing, gene-level quantification, quality control.** *Amphiprion ocellaris*. Raw (demultiplexed) fastq files were quality-checked based on reports generated by using FastQC v0.12.0 (RRID:SCR_014583; [140]) with default parameters, before and after adapter trimming. Adapters were trimmed using the function bbduk (RRID:SCR_016969) of BBTools v39.01 (RRID:SCR_016968; Bushnell B., http://sourceforge.net/projects/bbmap/) with ktrim=r, and k=23, trimpolyg=40. Categories flagged by FastQC after trimming ("warning" or "fail") were analyzed in detail and judged not to be prejudicial to further analysis. Issues relating to the detection of high sequence duplication rates were diagnosed based on the output of the analyzeDuprates function of the dupRadar package [141]. To this aim, duplicate reads were marked with the function markdup from SAMtools v1.18 (RRID:SCR_002105; [142]) with default parameters. As a further quality control step to detect sample contamination, trimmed reads were also run through FastQ Screen v0.15.2 (RRID:SCR_000141; [143]) with default parameters, against a manually curated set of genomes including—in addition to default ones—those of other fish species used by neighboring laboratories (goldfish *Carassius auratus*: ASM336829v1, carp *Cyprinus carpio*: ASM1834038v1, zebrafish *Danio rerio*: GRCz11), the three main live foods fed to our fish (*Artemia franciscana*: AFR02, *Brachionus rotundiformis*: ASM1680229v1, *Chlorella vulgaris*: cvul), and additional possible contaminating species (ant *Formica exsecta*: ASM365146v1). All FastQ Screen genomes were indexed with bowtie2 v2.4.4 (RRID:SCR_016368; [144]) as per FastQ Screen recommendation.

Trimmed reads were quantified at the transcript-level using the pseudo-aligner salmon v1.10.1 (RRID:SCR_017036; [145]) against the clownfish Amphiprion ocellaris reference transcriptome, using the genome as a decoy (decoy-aware pseudo-alignment; assembly ASM2253959v1; [146]), as per documentation, and using flags --validateMappings --seqBias

--gcBias. The average mapping rate was 85.9% of total reads. Salmon output transcript-based quantification files (quant. sf) were imported in RStudio (RRID:SCR_000432; [RStudio 147]) and summarized at the gene level using the tximport function from the tximport package (RRID:SCR_016752; [148]), referencing the gene models of the *Amphiprion ocellaris* reference genome assembly ASM2253959v1. The counts matrix was obtained by re-calculating counts through the flag countsFromAbundance = "lengthScaledTPM". Gene metadata (names, descriptions) were loaded from a custom-curated reference file based on the integration of Ensembl (still based on assembly AmpOce1.0) and NCBI annotations. This metadata reference is available at the code repository associated with this publication.

*Branchiostoma floridae*, *Ptychodera flava* and *Strongylocentrotus purpuratus.* The raw sequencing data of amphioxus, acorn worm and sea urchin were processed using the following bioinformatics pipeline. Initially, quality trimming was performed using *fastp* v0.22.0 to remove low-quality bases and adapter sequences [149]. The trimmed reads were then aligned to the reference genomes of *B. floridae* (GCF_000003815.2, NCBI), *P. flava* [150], and *S. purpuratus* (v5.0 genome assembly from Echinobase) using the STAR aligner v2.7.10b [151]. The STAR output files were subsequently sorted using SAMtools v1.17 to organize for downstream analyses [142]. Gene read counts were quantified using FeatureCounts v2.0.3 [152]. Finally, the count tables were processed with the DESeq2 v1.38.3 package in R for normalization and transformations, preparing the data for the downstream analysis [153].

*Platynereis dumerilii.* FastQC v. 0.11.9. [140] was used to quality control raw reads (fastq files) generated from sequencing. The raw reads were then preprocessed with Trimmomatic v. 0.39 to remove adapter sequences and low-quality regions, then passed through another round of quality control to verify successful adapter removal. Bases were removed from the start of each read if they had a Phred score below 20 (LEADING:20) and removed at the end of each read if they had a Phred score below 3 (TRAILING:3). A sliding window of 4 was used to continue removing bases based on an average score threshold of 15 (SLIDINGWINDOW:4:15). Any reads shorter than 30 bases after trimming were discarded, as they might be too short to provide reliable mapping (MINLEN:30) [154]. Alignment of raw reads to genome references was conducted by STAR 2.7.11a following [151] and the tabulation of aligned counts was carried out using the Subread featureCounts version 3 [152].

**Filtering and standard counts processing.**  Raw counts from each dataset were processed under analogous conditions, following the same processing pipeline. Specifically, counts were processed as a DGEobject (edgeR package, RRID:SCR_012802; [155–158]) and differences in library size were taken into account by obtaining counts per million (CPM) values (edgeR's function CPM) to allow a comparable threshold to filter out of lowly expressed genes. We only maintained genes for which at least 10 counts could be detected (arbitrary) in at least the number of samples corresponding to the number of replicates obtained from each species (i.e., in at least 3 or 4 samples). We note that the threshold of 10 counts, chosen a priori based on usual practices, resulted in the filtering out of key genes of interest (e.g., Wnt10 and Tbxt in Sp, Cdx4 in Ao). Though we later noticed that the pattern of expression of many filtered-out genes of interest is in fact coherent with expectation based on published adult gut literature or on known embryonic patterns, we preferred to prioritize robustness of results over expansion of the genes included in the comparison. We maintained the original count threshold set a priori.

Each filtered counts table was then processed through standard Deseq2 pipeline (DESeq2 package, RRID:SCR_015687; [153]) as in Mantica and colleagues [81] but further including VST normalization (DESeq2's varianceStabilizingTransformation function). Batch effects associated with the biological replicates of each dataset were removed using the removeBatchEffect function from the limma package (RRID:SCR_010943; [159]).

## Integrated analysis

**Orthogroup preprocessing.**  Count data from different species was integrated on the basis of Orthogroups, and as in Mantica and colleagues [81], unless indicated. Accordingly, and to decrease later computational complexity, Orthogroups were filtered by size and complexity: orthogroups comprising more than 120 genes were removed, as well as

orthogroups with more than 80 genes in cases where these had unbalanced representation across species. Of the filtered orthogroups, 6,118 were represented across all species (and could therefore be used for integrated analysis). To avoid having to discard 1-to-many orthologues, a situation exasperated by the inclusion of a teleost in our panel of species, we summarized multi-member groups in each species to a single "best ancestral orthologue", i.e., the ortholog in each species that most likely conserved the ancestral protein sequence. To do this, for each orthogroup, pairwise sequence similarity scored were calculated using the parSeqSim function from the protr package [160] with parameters type = "local", submat = "BLOSUM62", gap.opening = 2, gap.extension = 0.1, to recreate the mafft defaults used in Mantica and colleagues [81]. The protein with the highest average sequence similarity (among all sequence similarity scores to all proteins from all other species in the same orthogroup) was ultimately selected. Note that due to the computational intensity of this step in our system, we did not require the highest average sequence similarity score to have substantial margin from that of the second-best scoring protein. Also note that though the approach in Mantica and colleagues [81] furthers refines "best ancestral orthologue" assignment by factoring in conservation of expression profile (Pearson's expression correlations across matching tissues), we could not do this in our case since finding which gut segments are analogous across species was the matter of investigation in the first place.

**Integration of count matrices.** Comparative analyses were performed by integrating normalized count matrices from all five species into a single counts matrix spanning all samples, using orthogroups as rows/features (rather than genes) to provide a common term of comparison. This results in a necessary loss of all species-specific genes, and all genes belonging to orthogroups not expressed in all five adult guts. For "many-to-one" orthogroup relationships, which need to be resolved to one-to-one relationships, the "best ancestral orthologue" was selected as representative of the orthogroup, based on the criteria detailed above. Finally, integration was achieved by working on within-dataset z-score normalized counts (from each species; (tissue counts − species mean)/(sample standard deviation within species); base R's "scale" function with default parameters) rather than the original counts value themselves ([161]; as in Mantica and colleagues [81]). Within-species z-scoring has been consistently observed to be required to minimize inter-species variability, avoid sample clustering by species, and to allow samples to cluster by tissue [81,162]. This is also what we observed in our dataset.

### Data analysis and visualization

Pearson correlation matrices: pairwise Spearman's rank correlation coefficients between samples were calculated using the "cor" function (base R). PCA plots: PCA was computed on centered, unscaled counts (prcomp function, base R) and the samples' PC scores were plotted using ggplot2 (RRID:SCR_014601; [163]). Heatmaps: all heatmaps were plotted using the function Heatmap from the ComplexHeatmap package [164,165] reporting z-scores ((counts − mean)/sample standard deviation; base R's "scale" function with default parameters). To provide a better understanding of the overall expression level and effective range of expression of selected genes along the AP axis, their highest and lowest heatmap z-score values were further overlayed with the corresponding absolute transcripts per million.

### Rank 2 Elliptical (R2E) seriation

We stress that the investigation and recovery of patterns of gene expression in sets of samples with an intrinsic (ana-tomical) AP order revealed itself to be non-trivial. Built-in seriation methods associated with heatmap plotting packages (following hierarchical clustering) would invariably require additional subjective steps of manual re-ordering of the clus-ters obtained (branch flipping) to reconstitute an overall diagonal pattern of gene expression spanning the AP axis of the segments. We ultimately chose to adopt a reproducible, less-subjective seriation criterion able to highlight long-range gradients in expression pattern (Rank-2 Elliptical seriation, "R2E"; [63]) and used such a seriation criterion as an inves-tigation tool in of itself. Though R2E seriation is available in Rstudio as an ordination method prepackaged within the

"seriation" package [166], we manually implement it from first principles [63] to allow much more flexibility in our ability to study these patterns of expression, and to be able to easily obtain a seriation with correct directionality or with alternative starting points. Specifically, pearson correlation coefficient matrices between genes (or orthogroups, if working with the orthogroup-summarized matrix) were calculated iteratively, effectively calculating the pearson correlation coefficient between the previous pearson correlation coefficients at each successive round ("cor" function, base R), until reaching a correlation matrix of rank 2. The rank of each matrix was defined as the number of eigenvalues greater than $e^{-13}$. We generally found that 6 iterations were always sufficient to reach a rank 2 matrix given our datasets. As demonstrated in Chen [63] the left singular vectors of such a matrix fall on an ellipse in a two-dimensional subspace, and their relative position along such an ellipse can be used as a reordering solution for a Robinson matrix. Single value decomposition was thus applied to the first rank-2 matrix reached ("svd" function, base R) and the distribution of the left singular vector coordinates of each gene (svd$u) was visualized as an interactive plot and to easily identify the index of the first element of the series (usually corresponding to the element at the largest gap). This new, "R2E" order was obtained by extracting the clockwise or anticlockwise sequence of elements from this first starting point. The same approach—but using as starting point the correlation matrix between gut segments rather than between genes/orthogroups—was adopted for R2E seriation of gut segments where applicable.

### Unbiased TF analysis (SAMap)

To estimate the cross-species tissue similarity based on TF or non-TF expression we adopted SAMap [83] analysis pipeline by using bulk RNAseq dataset. The TFs and non-TFs gene count tables were imported by Seurat [167] in R and further converted to.h5ad files with SeuratDisk (https://github.com/mojaveazure/seurat-disk.git). Mapping scores were calculated using SAMap default setting.

### Model-based gene set analysis (MGSA)

To find enriched gene sets within (unranked) genes expressed in the adult gut tube of all five species, we relied on clownfish gene annotations and clownfish gene sets given our extensive experience with these annotations in this species (see also [168]), and the current much higher level of understanding of gene function in vertebrate species compared to the other species under analysis. We proceeded with the assumption that orthologous genes likely share functional properties. Accordingly, the clownfish best representative orthologue (see section "Integrated analysis") was taken as the representative of each expressed orthogroup, and Model-based gene set analysis (MGSA; mgsa function from mgsa package, [169]) was run by referencing the Ensembl "biological_process" Gene Ontology annotation for Amphiprion ocellaris ("aocellaris_gene_ensembl" dataset; useMart and getBM functions from biomaRt package, RRID:SCR_019214; [170,171]). Given that Ensembl annotations are based on a different (earlier) genome assembly (AmpOce1.0), orthogroups whose clownfish best representative orthologue had no matching Ensembl ID could not be taken into account.

**sPLS-DA.** To identify key discriminant variables of segment groups of interest, we applied the single-step cross-dataset integration and classification approach of the multivariate integration framework [88], combining within-dataset z-score normalization, and classification through sparse Partial Least Square Discriminant Analysis [172]. In sPLS-DA, datasets are decomposed in a way that does not maximize variance, as in PCA, but where instead the components retrieved from decomposition of each dataset are maximally covariant (with penalties applied for further feature selection) [173]. In using sPLS-DA, we take the approach modeled in Mantica and colleagues [81], in which its use is immediately possible given that tissues equivalences are known, thus ensuring that the markers identified through sPLS-DA are biologically grounded and informative. However, this technique could not be immediately deployed in our study, since the identification of equivalent gut segments was in fact one of the main tasks we needed to accomplish. We stress that, by design, sPLS-DA can find common markers to any arbitrary combination of tissues, even where these tissues are

not in fact equivalent, and the value of its application, in terms of biological insight, is therefore dependent on the actual biological equivalence between the segments chosen. It is for this reason that we first deploy unbiased, generalized explorative and classification approaches, and only then, having found, e.g., Hox coordinates and TF modules to ground our inferences of tissue equivalences, we use sPLS-DA to find discriminating markers of such segments.

## Supporting information

**S1 Fig. Dissection of the gastrointestinal tract in the 5 species considered, and segment collection.** For each species, representative pictures of the dissection process and subsectioning criteria to obtain the final gut segments as illustrated in Fig 1. A, anterior; P, posterior. Dashed lines with scissors, dissection boundary. Segments are numbered according to AP position, matching labels in Fig 1. **First row**: clownfish gut dissection. **Second row**: amphioxus gut dissection. **Third row**: acorn worm sections. **Fourth row**: sea urchin gut dissection. **Fifth row**: *Platynereis* gut dissection. (PNG)

**S2 Fig. Overview graphical summary of counts processing pipeline.**
(PNG)

**S3 Fig. R2E seriation of bilaterian adult gut segments based on the expression pattern of all conserved TFs. A)** Summary of the steps deployed to derive the final set of conserved, gut-expressed transcription factors (TFs) used for cross-species comparison. For each species, heatmap of the Spearman's Rank Correlation coefficients between segments, based on the expression of these same TFs. Segments are arranged according to R2E seriation, which matches AP position. **B)** Partial Generalized Association Plot summarizing TF expression data across bilaterian gut segments. Top: heatmap of the Spearman's Rank Correlation coefficients between all segments, regardless of species. Segments are ordered according to R2E seriation. Bottom: heatmap showing the expression pattern (z-scores) of all TFs considered. TFs (rows) are ordered according to R2E seriation-guided hierarchical clustering. Segments are color-coded by approximate equivalent AP position, as indicated in A. Solid green line indicates qualitative grouping of gut segments based on correlation patterns and known anatomical position. The data underlying this Figure can be found in https://doi.org/10.5281/zenodo.17746910.
(PNG)

**S4 Fig. PCA overview of conserved TF expression across bilaterian adult gut segments. A)** Distribution of all gut segments (all species) across the two main Principal Components (PCs; PC1, PC2) and based on the full set of conserved, gut-expressed transcription factors (TFs). Segments are color-coded by approximate equivalent AP position as in the legend provided on the right. Dashed vertical line indicates main separation between clusters. Left: Plot indicating the proportion of variance explained by each PC (scree plot). Blue dashed line: proportion of variance that would be explained if all components had equal contribution. **B)** Summary table of the top TFs associated with sample separation across PC1. Right/Left red highlight of the contribution scores indicate whether TF expression is driving samples to the right/left of the PC plot, respectively. **C)** Same data as in the central panel of A, but with segments color-coded by relative level of expression of each of the main PC1 drivers (as in B) and other TFs of interest (bottom row). Red, high; blue, low. Dashed vertical line indicates main separation between clusters. **D)** For each species, heatmap showing the expression pattern (z-scores) of the main drivers of separation across PC1. TFs (rows) are clustered according to hierarchical clustering. Solid and graded rectangles under the heatmaps indicate segments previously assigned to block and gradient gut compartments, as defined in previous sections. The data underlying this Figure can be found in https://doi.org/10.5281/zenodo.17746910.
(PNG)

**S5 Fig. Cross-species SAMap mapping of gut segments. A)** Heatmap of SAMap Mapping Scores between gut segments across species, based on the expression of all TF genes (upper triangle, above the diagonal) or all non-TF genes

(lower triangle, below the diagonal). In the SAMap approach, all paralogues are included for 1-to-many orthogroups. The data underlying this Figure can be found in https://doi.org/10.5281/zenodo.17746910.
(PNG)

**S6 Fig. Model-based Gene Set Analysis of non-TFs expressed across adult bilaterian gut segments. A)** Summary results of enriched Gene Ontology pathways across all gut-expressed non-transcription factor genes (Left) and among the top 2,500 most variant subset of them (Right). Pathways names in bold and highlighted in yellow have their expression pattern plotted in B. "in/tot": ratio between the number of genes of a given pathway present in the set considered, and the total number of genes belonging to that pathway; "estimate": MGSA score. **B)** For each species, heatmap showing the expression pattern (z-scores) of all expressed genes belonging to the GO pathway of interest. Genes (rows) are clustered according to hierarchical clustering. Segments (columns) are ordered according to AP position. Solid and graded rectangles under the heatmaps indicate segments previously assigned to block and gradient gut compartments, as defined in previous sections. The data underlying this Figure can be found in https://doi.org/10.5281/zenodo.17746910.
(PNG)

**S7 Fig. Module 2 sPLS-DA. A)** sPLS-DA of bilaterian gut segments marked by Module 2 TFs. Left: illustration of the segments to be discriminated (Module 2 segments). Right: separation of segments according to sPLS-DA discriminant genes. Bottom: For each species, heatmap showing the expression pattern (z-scores) of the top discriminant genes identified through sPLS-DA. Genes (rows) are clustered according to hierarchical clustering. Segments (columns) are ordered according to AP position. Solid and graded rectangles under the heatmaps indicate segments previously assigned to block and gradient gut compartments, as defined in previous sections. Gene names followed by an asterisk constitute the optimal sufficient discriminant set. Gene names in color are TFs. The data underlying this Figure can be found in https://doi.org/10.5281/zenodo.17746910.
(PNG)

**S8 Fig. Summary organization of each species gut tube. A)** Summary of the AP arrangement of compartments, modules, Hox and ParaHox genes, and other common functions (red segments) in the adult through-gut of each of the five bilaterian species investigated. **B)** Summary model of the likely ancestral configuration of the adult bilaterian through-gut, based on the patterns observed in extant species.
(PNG)

## Acknowledgments

We thank the High Throughput Sequencing Core hosted in the Biodiversity Research Center at Academia Sinica for performing all NGS experiments. The core facility is funded by Academia Sinica Core Facility and Innovative Instrument Project (AS-CFII-108-114). We further thank all the administrative staff and aquaculture specialists of Yilan Marine Research Station, and all present members of all four labs involved in this research project. We thank Paul Gerald Layague Sanchez for valuable input on this research and critical reading of this manuscript.

## Author contributions

**Conceptualization:** Stefano Davide Vianello, Vincent Laudet, Yi-Hsien Su, Jr-Kai Yu, Stephan Q Schneider.

**Data curation:** Stefano Davide Vianello, Ching-Yi Lin, Wahyu Cristine Pinem, Han-Ru Li, Kun-Lung Li.

**Formal analysis:** Stefano Davide Vianello, Ching-Yi Lin, Wahyu Cristine Pinem.

**Funding acquisition:** Vincent Laudet, Yi-Hsien Su, Jr-Kai Yu, Stephan Q. Schneider.

**Investigation:** Stefano Davide Vianello, Ching-Yi Lin, Wahyu Cristine Pinem, Han-Ru Li, Kun-Lung Li, Grace Sonia.

**Methodology:** Stefano Davide Vianello, Ching-Yi Lin, Wahyu Cristine Pinem, Han-Ru Li, Kun-Lung Li, Grace Sonia.

**Project administration:** Stefano Davide Vianello, Ching-Yi Lin, Wahyu Cristine Pinem, Han-Ru Li, Kun-Lung Li, Grace Sonia, Vincent Laudet, Yi-Hsien Su, Jr-Kai Yu, Stephan Q Schneider.

**Resources:** Stefano Davide Vianello, Ching-Yi Lin, Wahyu Cristine Pinem, Han-Ru Li, Kun-Lung Li, Grace Sonia, Shu-Hua Lee, Szu-Kai Wu.

**Software:** Stefano Davide Vianello, Ching-Yi Lin.

**Supervision:** Stefano Davide Vianello, Vincent Laudet, Yi-Hsien Su, Jr-Kai Yu, Stephan Q Schneider.

**Visualization:** Stefano Davide Vianello, Ching-Yi Lin.

**Writing – original draft:** Stefano Davide Vianello, Ching-Yi Lin, Wahyu Cristine Pinem, Han-Ru Li, Kun-Lung Li, Grace Sonia.

**Writing – review & editing:** Stefano Davide Vianello, Vincent Laudet, Yi-Hsien Su, Jr-Kai Yu, Stephan Q Schneider.

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
