## [Editor Report · Decision Letter 0]

22 Aug 2025

Dear Dr Vianello,

Thank you for submitting your manuscript entitled "Deconstructing the common anteroposterior organisation of adult bilaterian guts" for consideration as a Research Article by PLOS Biology.

Your manuscript has now been evaluated by the PLOS Biology editorial staff, as well as by an academic editor with relevant expertise, and I'm writing to let you know that we would like to send your submission out for external peer review.

Once your full submission is complete, your paper will undergo a series of checks in preparation for peer review. After your manuscript has passed the checks it will be sent out for review. To provide the metadata for your submission, please Login to Editorial Manager (https://www.editorialmanager.com/pbiology) within two working days, i.e. by Aug 26 2025 11:59PM.

Kind regards,

Roli Roberts

Roland Roberts, PhD

Senior Editor

PLOS Biology

rroberts@plos.org

---

## [Decision Letter · Decision Letter 1]

17 Oct 2025

Dear Dr Vianello,

Thank you for your patience while your manuscript "Deconstructing the common anteroposterior organisation of adult bilaterian guts" went through peer-review at PLOS Biology. Your manuscript has now been evaluated by the PLOS Biology editors, an Academic Editor with relevant expertise, and by three independent reviewers.

You'll see that reviewer #1 says that the paper is well-written and robust, but wants you to also present absolute expression levels, asks for more discussion of the divergence of the anterior-most segment, and thinks you should re-work and re-focus the Discussion. Reviewer #2 likes the study, but thinks that the manuscript is challenging to read, and the Results section needs to be more accessible to the broader reader. S/he has a long list of textual and presentational requests. Reviewer #3 says that the study is well designed, and the analysis is “thorough and careful,” but also complains about the writing being “convoluted and wordy,” suggesting that it should be substantially streamlined. S/he has a number of further technical questions and requests for discussion.

Overall, there's a clear consensus that your study is interesting and rigorous, but that the manuscript needs some significant presentational and textual improvement.

In light of the reviews, which you will find at the end of this email, we are pleased to offer you the opportunity to address the comments from the reviewers in a revision that we anticipate should not take you very long. We will then assess your revised manuscript and your response to the reviewers' comments with our Academic Editor aiming to avoid further rounds of peer-review, although we might need to consult with the reviewers, depending on the nature of the revisions.

**IMPORTANT - SUBMITTING YOUR REVISION**

*Resubmission Checklist*

*Published Peer Review*

*PLOS Data Policy*

*Blot and Gel Data Policy*

Sincerely,

Roli Roberts

Roland Roberts, PhD

Senior Editor

PLOS Biology

rroberts@plos.org

REVIEWERS' COMMENTS:

Reviewer #1:

[identifies himself as Chema Martin]

In "Deconstructing the common anteroposterior organisation of adult bilaterian guts", Vianello and co-authors investigate the anteroposterior patterning of bilaterian guts through bulk transcriptomic profiling. The evolution of a through-gut (a gut with a mouth and an anus) was, arguably, a transformative innovation that allowed the specialisation of the digestive system, potentially allowing the exploitation of new ecological niches and promoting speciation. Although most of the work has focused on investigating the formation and patterning of the through-gut in embryos and larvae, less is known about how adult bilaterian guts are patterned at the molecular level and whether that patterning is similar to that in embryos and conserved across lineages. To address this gap, the authors selected five representative species of more or less distantly related animal phyla (a vertebrate, an invertebrate chordate, a hemichordate, an echinoderm and an annelid) and dissected their adult guts into serial sections, which they then used to generate bulk transcriptomics in replicates. Applying sophisticated computational analyses, the authors discover that, despite significant transcriptomic differences, adult bilaterian guts are patterned into two primary regions (an anterior and a trunk/posterior region, primarily based on Hox gene expression) that can be further subdivided into four functional modules based on global transcriptomic signatures. Notably, genes commonly used to define endoderm/gut regions in bilaterian embryos and larvae are still deployed in the adults, generally maintaining the expected anteroposterior localisation based on the embryonic expression. This is seen even in the echinoderm, in which there is not a direct ontogenetic relationship between embryonic and adult guts. Overall, the manuscript is well-written, methodologies and results are clearly explained, and the observations support the main conclusions of the study.

I do not have major concerns about the analyses and approach, but I believe there are some points the authors may want to clarify:

- Most of the analyses are done at the level of z-scores, which is a good way to show and compare trends between genes, regions and species. However, it would also be meaningful to provide information on the actual expression levels, especially of transcription factors. For example, are Hox genes highly expressed in the adult gut? That is surprising, given that gene expression analyses in many invertebrates (at different life stages, including e.g. juveniles) have not detected a signal for these genes in the gut. The same applies to other TFs, as this would also help two align two seemingly contradictory results of the study: that gut transcriptomes differ significantly between species, yet there is some conserved signal, especially at the regulatory level (which, in all fairness, is not surprising).

- It is more interesting that the foremost segment (mouth? oral ectoderm?) is more divergent and, in the summary figure, is left out (i.e., there is no module in the anteriormost part of the through-gut). Could the authors discuss the implications of this finding further? This contrasts with the observations about the hindgut, where there is more conservation. Does this have any evolutionary implications?

- The discussion is long and very speculative. I do not think the authors' data provide new insights about the phylotypic stage question or the ancestrality of larval forms (actually, the alternative possibility, namely that an ancestral adult programme was deployed in the larvae as they evolved secondarily is not even considered and equally probable). The discussion about "the concept of the 'eigengene'" is confusing and is something definitely not mainstream in evo-devo. Likewise, statements like "AP shifts of these modules with respect to the fixed compartments defined by Hox genes would in turn be at the origin of evolutionary innovations. We see such shifts to be characteristic of the chordate branch." are not entirely substantiated by the data, which is a single transcriptomic snapshot with low evolutionary resolution (how do the authors know that the transcriptomic differences are at the base of the morphological differences in the gut and is not the other way around?). I suggest the authors rework the discussion to focus on the few main take-home messages and their clear implications.

Minor points:

- Generally, throughout the manuscript (text and figures), gene names should be in italics.

- In the discussion, the word adult is written in italics at some points. Is that on purpose?

- Figure 3: lacks a legend bar setting the range of z-scores in every panel. Annunziata et al is also in italics (not needed) but not gene names.

- Figure 4, 4x2, 5, 5x1, 6a, 6x, 6b: also lack legend bars for the heatmaps.

- Figure 7: in the top-left box below the A, there is a typo in gut (written as gutt)

Reviewer #2:

Vianello et al. present a molecular characterization of adult guts from five different species (P. dumerilii, S. purpuratus, P. flava, B. floridae, A. ocellaris) through anteroposteriorly-resolved bulk transcriptomics and cross-species comparisons to understand anteroposterior gene expression patterns and identify regional equivalences in gut segments across bilaterians, with the aim to unravel the evolution of gut patterning. The data generated is of good quality, presenting a solid approach to dissect an interesting topic such as gut patterning. The authors critically analyze the data, often highlighting the weak points of their approach and justifying their choices in terms of data analysis, which is commendable. However, the paper is dense in information, which makes it challenging to read and follow the authors' ideas between paragraphs. Moreover, the terminology used is not always well-suited to the concepts being discussed. We suggest improving some of the results sections as said in the next points, simplify the text and perhaps keep only the figures showing the essential analysis needed to support the conclusions, since some of the graphs seem unnecessary. Overall, the paper is worthy of publication, even though some re-writing and a review from a bioinformatic expert is recommended.

Major comments:

The results should be presented with an easier style, especially in their technical aspects, to make the paper more understandable for a general public, such as the audience of PLOS Biology. Also, line numbers and page numbers are missing, making the review difficult.

Minor comments:

Introduction, page 10: The authors should substitute:

"in the last common ancestor" with "in their last common ancestor",

"through-gut" ancestor with "through-gutted" ancestor,

"on a developmental biology basis" with "from a developmental biology perspective"

Page 13. "deeply homologous structures" with "yet homologous structures".

Results, Section 1, page 14: In this paragraph, the authors multiple times refer to anterior or posterior compartments without specifying the names of tracts that are a part of such compartments, e.g., "we note that i) anteriormost gut segments", " ii) segments posterior to this first domain". It would be better to clarify which gut segments (name the most anterior and posterior ones) form these molecular compartments.

Results, Section 3, page 22: "embrionic-" is "embryonic", remove "-"

Results, Section 3, Figure 3A-B: add color legend directly in the figure to make visualization easier, e.g. red/blue for expression levels (with values, where possible) and blue = foregut, yellow = midgut etc..

Results, Section 4, Figure 4D, substitute "to not match" with "not to match". The red squares underneath the heatmaps indicate the gut segments described in Figure 1, but even in this case it would be useful to write a legend to recall the reader of such fragments.

Results, Section 4, Supp. Figure 4x1: missing dot at the end of the sentence.

Results, Section 5, Figure 5A: a legend with expression values is missing.

Results, Section 5, Supp. Figure 5x1B: a legend with expression values is missing.

Results, Section 6, Figure 6: a legend with expression values is missing.

Page 15: the authors propose the "transition sphincter" as the constriction between the "block" and "gradient" components of an adult gut. At page 45, in Figure 7, they specify that the transition sphincter in adult bilaterian guts is located right anteriorly the Pdx+ domain, and they color the sea urchin larva's cartoon accordingly. Having to make a comparison between larva and adult, this "transition sphincter" would rather correspond to the so-called larval "cardiac sphincter" (the constriction between esophagus and stomach). If this is correct, in larvae (at least S. purpuratus 72hpf plutei) there is no Pdx domain right after that sphincter; Pdx1 transcripts are present only in correspondence of the pyloric sphincter, while Pdx1 protein expands up to mid-stomach, not in the complete stomach. Moreover, if once again this interpretation is correct, this figure is also misleading for the "Ex" domain: in the sea urchin larva, the exocrine domain is not present at the end of the "block" component. Therefore, for the figure not to be misleading, this reviewer suggests correcting the colors in the sea urchin larva cartoon, taking into consideration such differences.

Figure 7: Similarly to what described above, the anterior border of Pdx1 domain in panel A does not appear correct for sea urchin adults, because not in agreement with what reported in the heat map shown in Figure 3A. Therefore, also the color code reported for the sea urchin adult gut appears misleading.

In the Results, multiple times the authors refer to sPLS-DA and they write, at page 37: "Critically, our module-based interpretative framework allows us to impute shared segment identities across species, and therefore to apply supervised dimensionality reduction approaches such as Partial Least Squares - Discriminant Analysis (sPLS-DA, (Rohart et al., 2017)) to identify their discriminating markers and/or expand the signatures beyond TFs alone". For non-expert readers, could the authors add a brief justification, in the main text, of why this method is more suitable to the data they present? The same comment can be made for R2E representation.

Supplementary Figure 2x1 seems not to be essential for the general message of the paper, but can be relevant for clownfish specialists and Hox genes evolution-experts. If the authors want to keep such a figure, perhaps they could expand the paragraph in which they explain its meaning.

Page 22: "We show that the expression of all of these early patterning markers (including, here again, Hox and ParaHox genes) is conserved along the Strongylocentrotus purpuratus adult gut, virtually unchanged (Figure 3A)". The authors should also cite here the paper from Paganos et al., 2025, where the expression of the gut patterning genes was explored through snRNA-seq of a post-metamorphic stage (juvenile) of Paracentrotus lividus.

Page 22: "We find this conservation extraordinary, not only because it suggests the long-term permanence of a "larval" AP gut signature long into adulthood, but also because such signature is manifestly retained even across drastic ontogenic restructuring of the very physical materiality of the gut tube itself, as is the case of what happens specifically to the sea urchin gut tube during metamorphosis (Holland, 2020)". This sentence does not read well. Moreover, it could be more correct to propose that the regulatory program used in larval gut development is reused after metamorphosis for the patterning of the adult gut. The authors should also take into account that a part of the larval gut is retained during and after metamorphosis, becoming a part of the adult gut.

Page 22 "In other words, a subset of anteriormost or posteriormost gut markers, which would appear as having poor AP conservation, may be best understood at a more holistic pan-bilaterian level as "terminal" markers (see e.g. foxD, foxP, foxI, Figure 3B), and be conserved in this role across life stages and species". The word "terminal" is a bit misleading, as it is usually referring to terminal differentiation genes or to the end of a structure. The authors should find another way to describe them for their position in the stomach or cluster. Figure 3B allows for better elaboration. Instead of having a vague discussion of the "terminal" markers, they should modify this section properly, describing in detail what they see in every species and put it into an evolutionary scenario.

Page 25: remove "de facto", substitute with "as transcription factors".

Page 26: "deconstruct" instead of "decompose".

Brachyury is one of the most conserved markers for the posterior gut in many animals, it is also present in (Annunziata et al., 2014). Why is it not mentioned in the paper and excluded from the analysis?

Page 27: "This pattern echoes our previous recovery, at the species level, of a shared "terminal" pattern of gene expression, a conservation of gene expression at either terminus, and ontogenetic inter-termini shifts of gene expression at least in sea urchin." Same comment for "terminal" at page 22. The authors should rephrase it.

Reviewer #3:

The authors investigated the transcriptomic profile of five bilaterian species along its AP axis, one protostome (the annelid Platynereis dumerilii) and four deuterostomes.

They found that all species show an anterior block domain followed by a posterior gradient domain, verified by two bioinformatic approaches, PCA and heatmap.

The most likely organizer genes of this AP pattern are hox genes and indeed they find in all species that the hox genes show a colinear expression even if the Hox cluster itself has been disrupted in the genome. Importantly, they find (and partially confirm) that among the Parahox genes, gsx is barely expressed in the gut, while pdx and cdx consistently pattern the gradient domain of the bilaterian guts.

They also show that the conservation of gene expression pattern is also preserved for a list of transcription factors that were shown previously having a graded expression in the gut of the sea urchin (Annunziata et al., 2014).

Next they analysed all non-TF genes, which represent largely the effector and structural genes of the gut. Through the application of various bioinformatic tools, Again, they find fairly good conservation of AP patterning across species, especially between Platynereis and the two ambulacranian species. The two chordate species display significant rearrangements of the modules suggesting that these represent chordate innovations, consistent with other molecular and morphological evolutionary changes observed in vertebrates.

Given that the investigated species comprise protostomes and deuterostomes, this implies that this AP patterning system is an ancestral bilaterian trait. Remarkably, the authors find that much of the conserved molecular features are derived from maintenance of well conserved and studied embryonic regulators. This is intriguing given the drastic metamorphosis from a larva to the adult involving complete remodelling of most tissues. This strongly suggests that embryonic patterning genes are redeployed in adult guts and

maintain similar antero-posterior positioning, which has important implications on the long-standing question of whether the larva is ancestral or a secondary integration or whether the adult is an addition to a former life stage.

I find the study very well designed through the choice of the selected species (although a second protostome would have been desirable). While the experimental procedure is straight forward, the authors have undertaken major emphasis on the bioinformatic analyses of the data, testing and evaluating multiple ways to extract correlations of spatial gene expression patterns. This is done in a thorough and careful manner. The data represent an important resource for future investigations and the findings and conclusions have important implications for our understanding of the conservation of bilaterian body plans. I have only rather minor comments that should help to further improve the paper:

1) My major point is the writing. It is at many places convoluted and wordy. While the manuscript unfortunately does not contain any page or line numbers to point at specific positions, I counted nearly 19 pages of the Results part (after deducting the figures). While the discussion (here called Summary and conclusion) is shorter, it still could be more succinct. The current style is heavy on the reader and hence dilutes the interesting main messages. I strongly advise to drastically shorten the main manuscript and perhaps push part of the detailed result descriptions to the supplement.

2) The discussion is relatively short and overall is touching on a number of interesting questions, e.g. the question of whether the larva or the adult is ancestral. Along this line, I think they also could discuss their findings in the context of a related longstanding question in EvoDevo, i.e. whether the urbilaterian was simple (acoel worm like) or complex. Likewise, I find it interesting that the conservation is more pronounced among those protostome and deuterostome species that have a primary ciliary larva, compared to the direct developers in the two chordates. The authors are invited to speculate about the underlying cause for this correlation.

3) Some TF expression patterns appear to be less conserved between sea urchin and the other species. The authors suggest that they may rather be terminal markers, instead of anterior or posteriormost markers. Q: are they conserved among the others and is the sea urchin the outlier? If yes, is that perhaps linked to the inversion of part of the hox cluster within the genome?

4) Along the same line: Previous studies have shown by in situ hybridization that for instance FoxA2 and Bra (TbxT) are expressed in both proctodeum and stomodeum in sea urchins, sea stars and also Platynereis (e.g. Arendt et al., 2001). However, in the heatmap in Fig. 3A, these genes are detected only at one end. Is it possible that they have simply not be detected at the respective other end (or not as differentially expressed)? If yes, these genes (e.g. FoxD4, NeuroD) might be indeed terminal genes.

5) To make the important point that many of the genes patterning the adult gut represent embryonic developmental regulators that are maintained or redeployed (after sometimes drastic metamorphosis), they rely on the published data of the sea urchin larval gut (Annunziata et al., 2014). Are similar data also available for Platynereis or P. flava larva? An early study on Platynereis larvae (Arendt et al., 2001) points in this direction, but more recent single cell data might be useful in the regard. This could potentially strengthen the authors conclusions on this point.

6) Are there genes shared in their AP pattern among at least 2 species that are not found in sea urchin? In other words, is the sea urchin a good reference?

---

## [Editor Report · Decision Letter 2]

26 Nov 2025

Dear Dr Vianello,

Thank you for your patience while we considered your revised manuscript "Deconstructing the common anteroposterior organisation of adult bilaterian guts" for publication as a Research Article at PLOS Biology. This revised version of your manuscript has been evaluated by the PLOS Biology editors and the Academic Editor.

Based on our Academic Editor's assessment of your revision, we are likely to accept this manuscript for publication, provided you satisfactorily address the following data and other policy-related requests.

IMPORTANT - please attend to the following:

a) Please introduce an active verb into your Title to make it declarative. We suggest something like "Comparative transcriptomics reveal the common anteroposterior organisation of adult bilaterian guts"

b) Please declare the following competing interest: "YSH serves on the editorial board of PLOS Biology."

c) Please address my Data Policy requests below; specifically, we need you to supply the numerical values underlying Figs 1BCD, 2BC, 3AB, 4BD, 5AB, 6aAB, 6bAB, S3AB, S4ACD, S5, S6B, S7, either as a supplementary data file or as a permanent DOI’d deposition. I note that you already have an associated GitHub deposition (https://github.com/StefanoVianello/ASICOB_GCP_AdultBilaterianGuts). Please could you confirm whether the data and code in this deposition are sufficient to recreate the Figures? Also, because Github depositions can be readily changed or deleted, please make a permanent DOI’d copy (e.g. in Zenodo) and provide this URL (see below).

d) Please cite the location of the data clearly in all relevant main and supplementary Figure legends, e.g. “The data underlying this Figure can be found in S1 Data” or “The data underlying this Figure can be found in https://zenodo.org/records/XXXXXXXX

We expect to receive your revised manuscript within two weeks.

*Published Peer Review History*

*Press*

Sincerely,

Roli Roberts

Roland Roberts, PhD

Senior Editor

rroberts@plos.org

PLOS Biology

DATA POLICY:

Regardless of the method selected, please ensure that you provide the individual numerical values that underlie the summary data displayed in the following figure panels as they are essential for readers to assess your analysis and to reproduce it: Figs 1BCD, 2BC, 3AB, 4BD, 5AB, 6aAB, 6bAB, S3AB, S4ACD, S5, S6B, S7. NOTE: the numerical data provided should include all replicates AND the way in which the plotted mean and errors were derived (it should not present only the mean/average values).

CODE POLICY

DATA NOT SHOWN?

---

## [Editor Report · Decision Letter 3]

5 Dec 2025

Dear Dr Vianello,

Thank you for the submission of your revised Research Article "Comparative transcriptomics reveal the common anteroposterior molecular blueprint of adult bilaterian guts" for publication in PLOS Biology. On behalf of my colleagues and the Academic Editor, Andreas Hejnol, I'm pleased to say that we can in principle accept your manuscript for publication, provided you address any remaining formatting and reporting issues. These will be detailed in an email you should receive within 2-3 business days from our colleagues in the journal operations team; no action is required from you until then. Please note that we will not be able to formally accept your manuscript and schedule it for publication until you have completed any requested changes.

Sincerely, 

Roli Roberts

Senior Editor

PLOS Biology

rroberts@plos.org